# Contrastive Conditional–Unconditional Alignment for Long-tailed Diffusion Model

## Abstract

Training data for class-conditional image synthesis often exhibit a long-tailed distribution with limited images for tail classes. Such an imbalance causes mode collapse and reduces the diversity of synthesized images for tail classes. For class-conditional diffusion models trained on imbalanced data, we aim to improve the diversity and fidelity of tail class images without compromising the quality of head class images. We achieve this by introducing two simple but highly effective loss functions. Firstly, we employ an Unsupervised Contrastive Loss (UCL) utilizing negative samples to increase the distance/dissimilarity among synthetic images. Such regularization is coupled with a standard trick of batch resampling to further diversify tail-class images. Our second loss is an Alignment Loss (AL) that aligns class-conditional generation with unconditional generation at large timesteps. This second loss makes the denoising process insensitive to class conditions for the initial steps, which enriches tail classes through knowledge sharing from head classes. We successfully leverage contrastive learning and conditional-unconditional alignment for class-imbalanced diffusion models. Our framework is easy to implement as demonstrated on both U-Net based architecture and Diffusion Transformer. Our method outperforms vanilla denoising diffusion probabilistic models, score-based diffusion model, and alternative methods for class-imbalanced image generation across various datasets, in particular ImageNet-LT with $256 \times 256$ resolution.

## 1 Introduction

Recent advances in diffusion models (Ho et al., 2020; Song et al., 2020) have led to breakthrough in various generation tasks such as image generation (Zhang et al., 2023; Rombach et al., 2022; Ruiz et al., 2023), video generation (Ho et al., 2022a;b), image editing (Couairon et al., 2022; Kawar et al., 2023), 3D generation (Poole et al., 2022), etc. These diffusion-based generative models rely on large-scale datasets for training, which often follow a long-tailed distribution with dominant data for head classes and limited data for tail classes. Similar to the class-imbalanced recognition model (Liu et al., 2022) and the class-imbalanced generative adversarial network (Tan et al., 2020; Rangwani et al., 2022; 2023; Khorram et al., 2024), diffusion model is not able to generate high-quality images for tail classes due to the scarcity of training data. As shown in Fig. 1, the original diffusion model with transformer backbone  (Ma et al., 2024) generates inferior images for a tail class (red wine) when trained using a long-tailed version of ImageNet. We aim to increase the fidelity and diversity of tail class images while maintaining the quality of head class images for diffusion models. Our work has a positive societal impact on content generation of underrepresented groups.

We look into a highly related and crucial problem of long-tailed image recognition, for which contrastive learning is an effective method (Jiang et al., 2021; Li et al., 2022). Contrastive learning is a general learning paradigm of minimizing the distance of positive pairs while maximizing the distance of negative pairs, which has attracted lots of research interest for self-supervised representation learning (Chen et al., 2020). On the other hand, diffusion model has also emerged as an effective model for representation learning (Fuest et al., 2024; Baranchuk et al., 2021). We explicitly incorporate contrastive learning regularization to diffusion models to further enhance representation learning towards better image generation.

Our first loss is an unsupervised contrastive loss with negative samples only, serving as regularization for denoising diffusion probabilistic models (DDPM) and score-based diffusion models (SBDM).

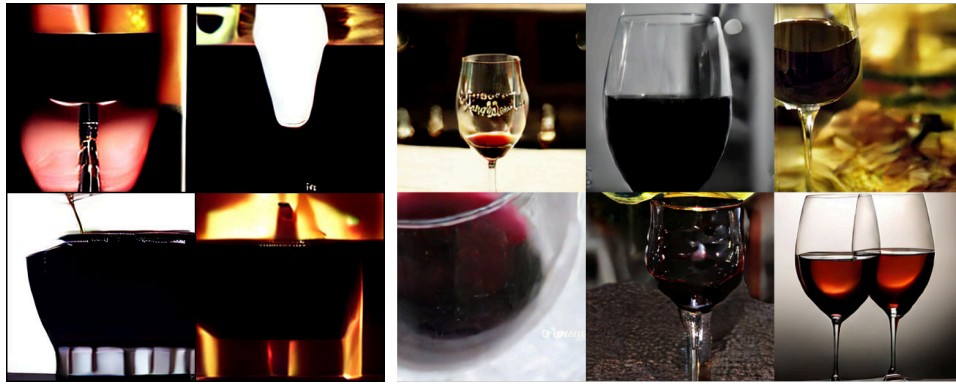

(a) SiT (Ma et al., 2024)          (b) SiT with CCUA (ours)

Figure 1: Synthetic images for a tail class ('red wine') by (a) standard SiT and (b) proposed CCUA framework. Both are trained on long-tailed ImageNet dataset for 160 epochs with 256x256 resolution.

Given that mode collapse during inference manifests as generated samples being overly similar for tail classes, we introduce unsupervised contrastive learning with negative samples to maximize the distances between generated images. Unlike supervised contrastive learning (Khosla et al., 2020), unsupervised contrastive learning distinguishes between representations of images regardless of their class, hence increasing intra-class and inter-class image diversity.

Our second loss is an alignment loss designed to align estimated noises from conditional and unconditional generation, which effectively minimizes the KL divergence between latent distributions for conditional and unconditional generation. While such conditional-unconditional alignment seems undesirable with less controllability, a critical aspect of our approach is that our alignment loss is weighted more for larger timesteps corresponding to the initial stage of the reverse process. Given the known overfitting in conditional generation for tail classes for both GAN, we are motivated by the success of conditional-unconditional alignment for long-tailed GAN (Khorram et al., 2024; Shahbazi et al., 2022), which facilitates knowledge sharing between head classes and tail classes. Notably, unconditional GAN generation has been observed to achieve superior FID than conditional generation under limited data (Shahbazi et al., 2022). We adapt conditional-unconditional alignment from GAN to diffusion models. Unlike the GAN-based method (Khorram et al., 2024) which aligns conditional generation and unconditional generation for low-resolution representations of images exhibiting intra-class similarity, we leverage observed image similarity during the initial denoising steps for tail classes and head classes (Si et al., 2024). Hence, we propose to match conditional generation with unconditional generation for large timesteps.

While contrastive learning has been proposed for diffusion models to improve adversarial robustness (Ouyang et al., 2023), find semantically meaningful directions (Dalva & Yanardag, 2024), accelerate training (Yu et al., 2024b), and regularize representation (Wang & He, 2025), we effectively leveraged contrastive learning for class-imbalanced diffusion models and demonstrate the superior performance of our method on long-tailed image generation via comprehensive experiments.

The main contributions of this work are as follows:

- We propose **C**ontrastive **C**onditional-**U**nconditional **A**lignment for Diffusion Model (**CCUA**) with imbalanced data. Our proposed contrastive loss with conditional-unconditional alignment are easy to implement with standard DDPM and SBDM pipeline using both UNet-based architecture and Diffusion Transformer (DiT).

- Our first loss, Unsupervised Contrastive Loss (UCL), employs unsupervised contrastive learning loss with negative samples only, enhancing within-class diversity.

- Our second Alignment Loss (AL) aligns unconditional generation and conditional generation for the initial steps in the denoising process, facilitating knowledge sharing between head and tail classes.

- We improved the diversity and fidelity of tail class images for conditional generation while maintaining the quality of head class images for multiple datasets and various resolutions, in particular ImageNet-LT with 256x256 resolution.

## 2 RELATED WORK

**Class-imbalanced Image Generation** Generative models such as VAE (Kingma et al., 2019), GAN (Karras et al., 2019; Brock et al., 2019), and diffusion models (Ho et al., 2020; Rombach et al., 2022) generate inferior images for tail classes when trained with real-world data with a long-tailed distribution. It has attracted a lot of research interest (Ai et al., 2023; Tan et al., 2020; Khorram et al., 2024; Qin et al., 2023; Zhang et al., 2024) to address this issue for various types of models. Many methods address the problem of class imbalance by augmenting training data for the tail classes. A VAE is fine-tuned on tail classes under a majority-based prior (Ai et al., 2023). It is observed that GAN (Tan et al., 2020) can amplify biases, leading to tail classes to be barely generated during inference, highlighting that the fairness in GAN needs improvement. Khorram et al. (2024) propose a GAN-based long-tailed generation method, named UTLO, which shares the latent representations of conditional GAN with unconditional GAN and implicitly shares knowledge between the head class and tail class. The motivation with UTLO is that low-resolution representations of images from GAN are similar for head classes and tail classes. We observe a similar phenomenon for denoised images in the initial steps of the denoising process, and further propose a conditional-unconditional alignment loss designed for diffusion models. Recent work addresses class-imbalanced diffusion models (Qin et al., 2023; Yan et al., 2024; Zhang et al., 2024) by regularization losses to align or separate the distributions of synthetic images and their corresponding latent representations across different classes. For example, CBDM (Qin et al., 2023) loss minimizes the distance of estimated noise items between these two models, the original DDPM model and a second model trained with pseudo labels which form a uniform distribution. DiffROP (Yan et al., 2024) attempts to combine contrastive learning with diffusion model by maximizing the distance of distributions between classes. However, DiffROP only considers pairs of images of different classes as negative pairs without regularizing images of the same class. Conceptually, the first part of our method, the unsupervised contrastive loss designed to disperse unconditional latent space, generalizes DiffROP by including extra negative pairs of images within each class, hence promoting intra-class diversity.

**Contrastive Learning for Representation Learning** Both unsupervised contrastive learning (Chen et al., 2020; He et al., 2020) and supervised contrastive learning (Khosla et al., 2020; Li et al., 2022) are approaches used in representation learning to produce feature embeddings by maximizing the similarity between positive pairs and minimizing the similarity between negative pairs. Unsupervised contrastive learning does not require labels and typically augments the same data to form positive pairs (Chen et al., 2020; He et al., 2020). Supervised contrastive learning incorporates class labels and pulls together all samples from the same class while pushing apart samples from different classes. Contrastive learning is an effective method for self-supervised learning (He et al., 2020) and long-tailed recognition (Jiang et al., 2021; Li et al., 2022). We effectively utilized contrastive learning for class-imbalanced diffusion models. Our work is different from the contrastive-guided diffusion process (Ouyang et al., 2023), which aims to improve adversarial robustness. Another difference is that we focus on contrastive learning during training, while contrastive-guided diffusion process (Ouyang et al., 2023) is about guidance for inference. While the diffusion model originally for generation tasks becomes an emerging technique for representation learning (Fuest et al., 2024; Baranchuk et al., 2021), we directly integrate contrastive learning regularization to diffusion models for better generation on imbalanced data. Notably, unsupervised contrastive loss with negative samples exclusively focuses on separating dissimilar instances in the embedding space, which we leverage to address mode collapse, as detailed in Sec . 3.1. In the context of generative modeling, REPA (Yu et al., 2024b) aligns diffusion model features with features from a frozen vision encoder to accelerate training. Unlike REPA, we directly regularize diffusion model features without external models. Most relevant is concurrent work of dispersive loss (Wang & He, 2025), which is similar to our unsupervised contrastive loss with negative pairs only. Dispersive loss is directly and explicitly applied to conditional training, as designed for balanced diffusion models. By comparison, our unsupervised contrastive loss is formulated in the unconditional latent space and is implicitly extended to conditional training through our conditional–unconditional alignment loss. This carefully designed contrastive conditional-unconditional alignment framework for long-tailed diffusion models achieves better generation as shown in experiments.

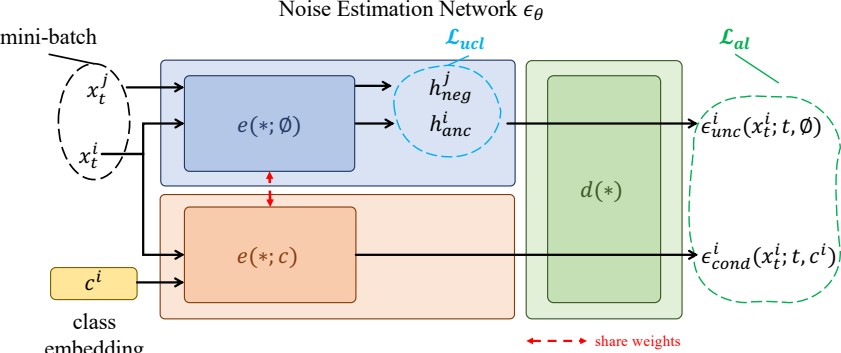

Figure 2: Overview of the proposed CCUA framework of contrastive learning for diffusion model. The noise estimation network is divided into two parts, a latent encode network $e(*)$ and a decode network $d(*)$. $e(*)$ encodes an image with noise $x_t$ to a low-dimensional latent $h$, which is decoded to noise $\epsilon$ by $d(*)$. Specifically, the latent encode network $e(*)$ and decode network $d(*)$ for unet-based model and diffusion transformer are shown in Fig. 5 in the appendix.A.1. We increase the distance of negative pairs of unconditional encoded latents for different samples by an unsupervised contrastive loss $\mathcal{L}_{ucl}$ with negative samples only, and align unconditional and conditional generation for the same sample $x_t^i$ and utilize an alignment loss $\mathcal{L}_{al}$ to minimize their distances at initial time steps.

## 3 METHOD

We propose **C**ontrastive **C**onditional-**U**nconditional **A**lignment (CCUA) framework for Diffusion Model. Fig. 2 gives an overview of our framework, while Fig. 5 in the appendix A.1 details how we apply CCUA framework into UNet-based model and Diffusion Transformer. Our framework involves a unsupervised contrastive loss with negative pairs only, and a conditional-unconditional alignment loss, as outlined in Sec. 3.1 and Sec. 3.2, respectively. Sec. 3.3 summarizes our overall framework.

### 3.1 UNSUPERVISED CONTRASTIVE LOSS

We observed the synthesized images of DDPM for a tail class concentrated around the limited training images in the latent space, leading to the issue of mode collapse, as shown in the Fig. 9, Fig. 10 in the appendix A.4.3, and Fig. 8 in the appendix A.4.2 with visualizing the distribution of synthetic images for the tail class. By applying the contrastive loss with negative samples only, we intend to increase the distance of each synthesized image from other images in the latent space, hence increasing the diversity of synthetic images in particular for tail classes.

We consider contrastive loss with negative samples only, where the negative samples are based on noisy images from a mini-batch. Specifically, we divide a noise estimation network into two parts, a latent encode network $e(*)$, and decode network $d(*)$. $e(*)$ encodes an image with noise $x_t$ to a low-dimensional latent $h$, which is decoded to the estimate noise item $\epsilon$ by $d(*)$. Our contrastive loss is defined for the latents $h$ with each image in a mini-batch treated as an anchor. All other images are treated as negative samples. The anchor sample $x_t^i$ and negative samples $x_t^j$ are fed into the encoder $e(*)$ to get the latents $h_{anc}^i = e(x_t^i; \emptyset)$ and $h_{neg}^j = e(x_t^j; \emptyset)$. Our unsupervised contrastive loss $\mathcal{L}_{ucl}$ is defined as follows:

$$\mathcal{L}_{ucl} = -\frac{1}{|B|} \sum_{i \in B} \log \frac{\pi_{anc}^i}{\pi_{anc}^i + \sum_{j \in B, j \neq i} \pi_{neg}^j}, \text{with } \pi_{anc}^i = \exp(\frac{h_{anc}^i \cdot h_{pos}^i}{\tau}), \pi_{neg}^j = \exp(\frac{h_{anc}^i \cdot h_{neg}^j}{\tau}),$$

(1)

where $B$ denotes a mini-batch, and $\tau$ is a temperature for softmax which we keep $0.1$ as the default setting. Note that it is common to augment an anchor image $x_t^i$ to be a positive sample in many unsupervised contrastive learning methods. Here, we don't apply any data augmentation and directly treat the anchor latent $h_{anc}^i$ itself as the positive vector in the contrastive loss, i.e., $h_{pos}^i := h_{anc}^i$.

**Why contrastive loss encourages diversity and discourages mode collapse?** In the worst case, that all embeddings collapse to a constant vector, the contrastive loss becomes $\log N$ for a batch with $N$ samples. Such constant embeddings yield the maximum possible loss and are therefore discouraged. Intuitively, contrastive loss introduces a repulsive force between different samples. The numerator remains constant, as self-similarity is always maximal, while the denominator aggregates pairwise similarities across the batch. Minimizing the loss requires reducing similarities between distinct samples, effectively pushing their embeddings apart in feature space. In this way, the model avoids collapse by maximizing inter-sample distances. Geometrically, the optimal solution (in the absence of augmentation) corresponds to embeddings being uniformly distributed on a hypersphere.

**Batch Resample** is a simple strategy for long-tailed recognition and generation (Zhang et al., 2024). However, it may lead to mode collapse due to repetitive images for tail classes. Since our unsupervised contrastive loss relies on pairs of images, batch resampling increases the chance of images from the same tail class appearing in the same batch, which further diversify tail class images. We choose batch resampling as an optional strategy for our framework and provide an ablation study in Sec. 4.3.

## 3.2 CONDITIONAL-UNCONDITIONAL ALIGNMENT LOSS

**Conditinal-unconditinal Alignment Facilitates Knowledge Sharing for Class-imbalanced GAN** We take inspiration from UTLO (Khorram et al., 2024), which addresses long-tailed generation with GANs, and observes that the similarity between head class and tail class images increases at lower resolution representations. To share knowledge between the head class and tail class, UTLO (Khorram et al., 2024) proposes unconditional GAN objectives for lower-resolution representations and conditional GAN objectives for subsequent higher-resolution images. More specifically, the lower-resolution representations becomes less sensitive to class labels, and the second part of the generator generates the fine details of images. Utilizing unconditional generation for lower resolution is effective in increasing the diversity and quality of tail class images. In a different setting of training GAN with limited data, Transitional-GAN (Shahbazi et al., 2022) leverages unconditional generation in a two-staged training scheme, as unconditional generation can give a better FID than conditional generation when data is limited.

**Conditional-unconditinal Alignment for Class-imbalanced Diffusion Model** We aim to adapt the conditional-unconditional alignment approach used in GANs (Khorram et al., 2024; Shahbazi et al., 2022) to diffusion models. Our key insight is that the diffusion model denoises images recursively, and the similarity between head class images and tail class images is higher during the *initial timesteps* of the denoising process. This is similar but different from UTLO (Khorram et al., 2024), which proposes unconditional generation for *lower-resolution* representations. We visualize the reverse processing of unconditional generation and conditional generation with different class labels of the original DDPM, starting from the same Gaussian noise, as shown in Fig. 3 and more examples in appendix A.4.4. Images from unconditional generation and conditional generation share similar low-frequency components, e.g., the coarse shape of synthesized objects and background region, especially at the initial time steps (Si et al., 2024).

Motivated by this observation, we propose to match unconditional generation and conditional generation at the initial time steps, to enable knowledge sharing between tail classes and head classes with abundant data. Another motivation is that class conditions are not necessary for all timesteps. In fact, using adaptive guidance weight with a small weight in the initial steps can yield a better FID compared to using constant guidance weight (Wang et al., 2024).

**From Conditional-unconditional Distribution Matching to Alignment Loss** We penalize KL divergence between the conditional distribution $p_\theta(x_{t-1}|x_t^i, c^i)$ and unconditional distribution $p_\theta(x_{t-1}|x_t^i)$:

$$\mathcal{L}_{al}^{i,t} = D_{KL}[p_\theta(x_{t-1}|x_t^i, c^i)||p_\theta(x_{t-1}|x_t^i)]. \tag{2}$$

Suppose $p_\theta(x_{t-1}|x_t, c) = \mathcal{N}(x_{t-1}; \mu_\theta(x_t, t, c), \sigma_t^2 I), p_\theta(x_{t-1}|x_t) = \mathcal{N}(x_{t-1}; \mu_\theta(x_t, t), \sigma_t^2 I)$, then we have:

$$\mathcal{L}_{al}^{i,t} = \mathbb{E}[\frac{1}{2\sigma_t^2}||\mu_\theta(x_t^i, t, c^i) - \mu_\theta(x_t^i, t)||^2] + C, \tag{3}$$

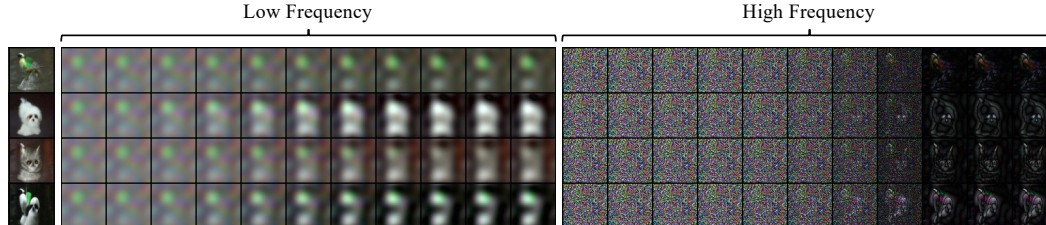

Figure 3: The leftmost column shows synthetic images with different classes but with the same initial noise or random seed. We visualize the low-frequency component and the high-frequency component from the reverse processing. It shows that low-frequency components are similar during initial time steps for different classes with the same initial noise (Si et al., 2024). More examples are in appendix A.4.4.

where C is constant, and

$$\mu_\theta(x_t^i, t, c^i) = \frac{1}{\sqrt{\alpha_t}}x_t^i - \frac{\beta_t}{\sqrt{\alpha_t}\sqrt{1-\bar{\alpha}_t}}\epsilon_\theta(x_t^i, t, c^i), \quad \mu_\theta(x_t^i, t) = \frac{1}{\sqrt{\alpha_t}}x_t^i - \frac{\beta_t}{\sqrt{\alpha_t}\sqrt{1-\bar{\alpha}_t}}\epsilon_\theta(x_t^i, t),$$

(4)

where $\alpha_t = 1 - \beta_t$, $\bar{\alpha}_t = \prod_{i=1}^{t}(1-\beta_i)$, $\{\beta_t\}_{1:T}$ is the variance schedule. Then, $\mathcal{L}_{al}$ simplifies to:

$$\mathcal{L}_{al}^{i,t} \propto ||\epsilon_\theta(x_t^i, t, c^i) - \epsilon_\theta(x_t^i, t)||^2.$$

(5)

As shown in Fig. 2, the anchor vector $h_{anc}^i$ is fed into decode network $d(*)$ to get unconditional noise estimation $\epsilon_{unc}^i := \epsilon_\theta(x_t^i; t, \emptyset)$. Meanwhile, the anchor image $x_t^i$ is also incorporated with the class label condition $\mathbf{c^i}$ fed into the network to get the conditional noise estimation $\epsilon_{cond}^i := \epsilon_\theta(x_t^i; t, \mathbf{c^i})$. The alignment loss is defined between the unconditional and conditional noise estimation:

$$\mathcal{L}_{al} = \frac{1}{|B|}\sum_{i \in B}\mathbb{E}_{t,x_0}[\frac{t}{T}||\epsilon_\theta(x_t^i; t, \mathbf{c^i}) - \epsilon_\theta(x_t^i; t, \emptyset)||^2].$$

(6)

The loss is weighted linearly by timesteps $t$, so that the initial steps with large $t$ are weighted more. In other words, we align conditional generation and unconditional generation for the initial steps.

### 3.3 OVERALL FRAMEWORK

Our final loss function $\mathcal{L}$ is the sum of the standard DDPM (Ho et al., 2020) loss $\mathcal{L}_{ddpm}$ and our contrastive conditional-unconditional alignment loss $\mathcal{L}_{ccua}$:

$$\mathcal{L}_{ccua} = \alpha \cdot \mathcal{L}_{ucl} + \gamma \cdot \mathcal{L}_{al}, \quad (7)$$

where $\alpha$ and $\gamma$ are the hyper-parameters for our unsupervised contrastive loss and alignment loss.

The overall algorithm framework is shown as Algorithm 1. In classifier-free guidance (CFG), the class labels $\mathbf{c}$ are randomly dropped by a specific probability $p_{uncond}$. In our setting, we keep the $p_{uncond} = 10\%$, same as (Qin et al., 2023). The parameters of the noise estimation network $\epsilon_\theta$ are updated by the gradient of the final loss $\nabla_\theta \mathcal{L}$ as in Eq. 7.

---

**Algorithm 1** Training algorithm of **CCUA**.

Set $\mathcal{L}_{ddpm}, \mathcal{L}_{ucl}, \mathcal{L}_{al} = 0$
**for** each image-class pair $(x_0^i, c^i)$ in this batch $B$ **do**
    Sample $\epsilon^i \sim \mathcal{N}(\mathbf{0}, \mathbf{I})$, $t \sim \mathcal{U}(\{0, 1, ..., T\})$
    $x_t^i = \sqrt{\bar{\alpha}_t}x_0^i + \sqrt{1-\bar{\alpha}_t}\epsilon^i$
    $\epsilon_{unc}^i = \epsilon_\theta(x_t^i; t, \emptyset), \epsilon_{cond}^i = \epsilon_\theta(x_t^i; t, c^i)$
    $h_{anc}^i = e(x_t^i)$
    $\pi_{anc} = \exp(h_{anc}^i \cdot h_{anc}^i/\tau)$, Set $\pi_{neg} = 0$
    **for** $x_t^j$ in this batch with $i \neq j$ **do**
        $h_{neg}^j = e(x_t^j)$
        $\pi_{neg} = \pi_{neg} + \exp(h_{anc}^i \cdot h_{neg}^j/\tau)$
    **end for**
    $\mathcal{L}_{ucl} = \mathcal{L}_{ucl} + (-\log\frac{\pi_{anc}}{\pi_{anc}+\pi_{neg}})$
    **if** Unconditional Training of CFG **then**
        $\mathcal{L}_{ddpm} = \mathcal{L}_{ddpm} + ||\epsilon^i - \epsilon_{unc}^i||^2$
    **else if** Conditional Training of CFG **then**
        $\mathcal{L}_{ddpm} = \mathcal{L}_{ddpm} + ||\epsilon^i - \epsilon_{cond}^i||^2$
        $\mathcal{L}_{al} = \mathcal{L}_{al} + \frac{t}{T}||\epsilon_{unc}^i - \epsilon_{cond}^i||^2$
    **end if**
**end for**
$\mathcal{L} = \frac{1}{|B|}(\mathcal{L}_{ddpm} + \alpha \cdot \mathcal{L}_{ucl} + \gamma \cdot \mathcal{L}_{al})$

---

**Synergy between unsupervised contrastive loss and alignment loss** Our proposed two losses including unsupervised contrastive loss $\mathcal{L}_{ucl}$ and $\mathcal{L}_{al}$ are novel by themselves in the context of diffusion models. What's more, the synergy of the two losses makes our method effective. Our $\mathcal{L}_{ucl}$ loss diversifies *uncoditinal* latents repulsing negative pairs, while $\mathcal{L}_{al}$ aligns *conditional* and *unconditional* latents/representations. As a result, conditional latents are diversified implicitly.

We choose not to directly diversify conditional latents, as is done in concurrent work of Dispersive Loss (Wang & He, 2025). Our combined loss is more effective for addressing overfitting in tail classes empirically verified in Tab. 1. The geometric interpretation for the reason is that the diversity of conditional latents involves both inter-class variance and intra-class variance. Directly applying InfoNCE loss on conditional latent (Wang & He, 2025) can lead to a shortcut solution that dominantly maximizes inter-class variance. However, contrastive regularization on unconditional latents *must* diversity latents for all data regardless of class. Aligning conditional latents to diversified unconditional latents via distillation is more effective. In Appendix A.5, we give a theoretical analysis through mutual information between data and latents for InfoNCE objectives. Unlike dispersive loss that focuses on conditional generation only, our method improves both conditional and unconditional generation, which are combined in classifier-free guidance during inference.

## 4 EXPERIMENTS

Table 1: Comparison on ImageNet-LT $256 \times 256$ with SiT pipeline. We use parentheses () to highlight the improvement of our method over SiT baseline on FID and Recall, which denotes the overall quality and diversity, respectively. Our method dramatically improves $\text{FID}_{tail}$ in particular.

| Epochs | Method | IS ↑ | FID ↓ | sFID ↓ | Prec. (%) ↑ | Recall (%) ↑ | $\text{FID}_{head}$ ↓ | $\text{FID}_{tail}$ ↓ |
|---|---|---|---|---|---|---|---|---|
| 40 | SiT | 53.9 | 33.8 | 22.6 | 54.5 | 19.1 | 23.3 | 52.8 |
| | CBDM | 54.8 | 34.1 | 23.3 | 53.9 | 18.7 | 23.3 | 53.1 |
| | REPA | **74.1** | 28.4 | 20.0 | 58.3 | 16.1 | **17.8** | 48.5 |
| | Dispersive Loss | 53.8 | 34.0 | 22.6 | 54.8 | 19.8 | - | - |
| | **CCUA (ours)** | 73.6 | **25.8** (-8.0) | **16.5** | **58.6** | **23.1** (+4.0) | 21.5 | **28.1** |
| 80 | SiT | 78.8 | 25.7 | 21.9 | 64.3 | 18.5 | 18.5 | 41.6 |
| | CBDM | 84.0 | 24.7 | 21.8 | 64.6 | 18.7 | 18.2 | 40.7 |
| | REPA | **105.9** | 21.9 | 19.5 | 65.7 | 15.2 | **17.0** | 39.7 |
| | Dispersive Loss | 83.8 | 25.2 | 21.7 | 65.5 | 17.3 | - | - |
| | **CCUA (ours)** | 103.6 | **19.9** (-5.8) | **14.8** | **65.8** | **21.3** (+2.8) | 17.1 | **21.9** |
| 120 | SiT | 103.1 | 21.2 | 20.1 | 69.3 | 18.2 | 17.5 | 35.4 |
| | CBDM | 105.7 | 20.94 | 20.7 | **70.3** | 17.8 | 17.7 | 35.5 |
| | REPA | **126.8** | 19.7 | 20.3 | 68.0 | 15.8 | 17.7 | 36.9 |
| | Dispersive Loss | 104.0 | 21.3 | 20.4 | 68.0 | 18.5 | **17.0** | 35.9 |
| | **CCUA (ours)** | 119.1 | **17.5** (-3.7) | **13.8** | 68.3 | **21.2** (+3.0) | 17.4 | **20.0** |
| 160 | SiT | 111.7 | 19.9 | 20.1 | 70.3 | 18.6 | 17.8 | 33.9 |
| | CBDM | 117.2 | 19.4 | 20.2 | **72.5** | 17.7 | 17.9 | 32.7 |
| | REPA | **137.8** | 18.1 | 19.3 | 69.8 | 16.2 | 18.0 | 33.8 |
| | Dispersive Loss | 115.6 | 19.7 | 20.1 | 69.9 | 18.9 | **17.0** | 34.1 |
| | **CCUA (ours)** | 124.8 | **16.4** (-3.5) | **13.1** | 69.8 | 21.3 (+2.7) | 17.8 | **19.5** |
| | **CCUA (ours)**[1] | 119.0 | 17.2 (-2.7) | 13.9 | 68.4 | **21.6** (+3.0) | 17.5 | 20.1 |
| 240 | SiT | 132.2 | 17.8 | 19.6 | **74.4** | 17.6 | 18.8 | 29.6 |
| | **CCUA (ours)** | **141.8** | **14.6** (-3.2) | **13.2** | 73.5 | **19.6** (+2.0) | **18.5** | **21.6** |

### 4.1 EXPERIMENTAL SETUP

We report results on multiple datasets, including the long-tailed versions of ImageNet, TinyImageNet, and Places datasets, namely ImageNet-LT, TinyImageNet-LT, and Places-LT, which are commonly used datasets in long-tailed image generation and recognition. We also report results and conduct more analysis on CIFAR10-LT/CIFAR100-LT datasets, which are shown in the appendix A.3. We measure IS Score, FID, spatial FID (Dhariwal & Nichol, 2021), Kernel Inception Distance (KID) (Bińkowski

---

[1]Our method incurs longer training time as discusseded in Sec. 4.4. For fair comparison, we also report results of model trained with our method for the same training time with SiT baseline (160 epochs for SiT).

Table 2: Comparison on TinyImageNet-LT $64 \times 64$ and Places-LT $64 \times 64$ with DDPM pipeline. Blue '()' shows improvement of our method over DDPM baseline. Green '()' shows improvement of DDPM trained on balanced version over trained on long-tailed version.

| Dataset | Method | FID↓ | FID$_{tail}$ ↓ | KID$_{\times 1k}$ ↓ |
|---|---|---|---|---|
| TinyImageNet-LT | DDPM$^*_{bal}$ (Ho et al., 2020) | 15.73 (-2.94) | 25.62 (-12.26) | 3.19 (-3.12) |
|  | DDPM (Ho et al., 2020) | 18.67 | 40.12 | 6.31 |
|  | CBDM (Qin et al., 2023) | 20.90 | 48.07 | 6.55 |
|  | OCLT (Zhang et al., 2024) | 17.72 | 39.67 | 5.61 |
|  | **CCUA (ours)** | **15.24** (-3.43) | **30.39** (-9.73) | **3.83** (-2.48) |
| Places-LT | DDPM (Ho et al., 2020) | 13.89 | 23.74 | 5.26 |
|  | CBDM (Qin et al., 2023) | 15.15 | 26.07 | 5.64 |
|  | OCLT (Zhang et al., 2024) | 13.04 | 22.84 | 4.17 |
|  | **CCUA (ours)** | **11.99** (-1.90) | **20.84** (-2.90) | **3.63** (-1.63) |

et al., 2018) as the overall quality metrics. We also report FID of tail classes (FID$_{tail}$), inclined to measure quality for tail classes. Precision and Recall are reported to respectively measure fidelity and diversity, which are widely used in evaluating generative models on long-tailed scenes (Qin et al., 2023; Zhang et al., 2024; Khorram et al., 2024). More implementation details are provided in the appendix A.1. We compare our proposed methods with multiple baselines, covering diffusion transformer and unet-based architecture. For diffusion transformer, we compare CCUA to SiT (Ma et al., 2024) and Dispersive Loss (Wang & He, 2025). For unet-based architecture, we compare to the original DDPM (Ho et al., 2020), CBDM (Qin et al., 2023), and OCLT (Zhang et al., 2024).

## 4.2 Quantitative Results

**Class-imbalanced Generation for Diffusion Transformer** We apply the proposed CCUA loss into the standard diffusion transformer training pipeline, SiT (Ma et al., 2024), on ImageNet-LT dataset, and compare it to Dispersive Loss (Wang & He, 2025), a concurrent work of applying contrastive learning for improving SiT on balanced dataset. We also compare to CBDM (Qin et al., 2023), a long-tailed training method for UNet-based model, and REPA (Yu et al., 2024a), the latest training technique for general DiT/SiT-based model. All models are sampled with ODE flow 50 steps for conditional generation with CFG strength 7.5. As shown in Table 1, our method achieves remarkable improvement compared to SiT, CBDM, REPA and Dispersive Loss on various training epochs. Our method achieves above 10% improvement on Recall, while keep the similar Precision as SiT, illustrating the higher diversity of the proposed CCUA loss without trading off fidelity.

**Class-imbalanced Generation for DDPM** For training unet-based architecture DDPM on TinyImageNet-LT and Places-LT datasets, our method also achieves the best performance, as shown in Tab. 2. All models are sampled with DDIM 100 steps for conditional generation with optimal CFG strength. For TinyImageNet-LT datasets, we provide the metrics of the DDPM model trained on the balanced version, i.e., the original TinyImageNet datasets, denoted by DDPM$^*_{bal}$, as the theoretical optimal reference. The best FID and KID score of our method illustrates the high quality of images synthesized by our method[2].

**Results Across Various Category Intervals** To see the effectiveness of our method on tail classes, we divide the Places-LT and TinyImageNet-LT datasets into three super categories: 'Head', 'Body', and 'Tail', with classes sorted by the number of training images. For each dataset, the top 33% classes were allocated to the 'Head' category, the next 34% classes to the 'Body' category, and the last 33% classes to the 'Tail' category. The percentage $P_{category}$ of training images belonging to each category is shown in the first row of Tab. 3. Our method achieves the best FID score across all three categories on Places-LT and TinyImageNet-LT datasets. For the 'Tail' classes, which are only $\sim 3\%$, our method improves their FID from **23.74** to **20.84** on Places-LT and **40.12** to **30.39** on TinyImageNet-LT, respectively. Additionally, both 'Head' and 'Body' classes show improved FID scores with our method, as demonstrated in Tab. 3.

---

[2]The FID score of our reproduced DDPM, CBDM (Qin et al., 2023), and OCLT (Zhang et al., 2024) are better than the numbers reported in (Qin et al., 2023; Zhang et al., 2024). We train all methods from scratch without fine-tuning and find the optimal guidance strength $\omega$ for each method. Such implementation is different from the one used in (Qin et al., 2023; Zhang et al., 2024) but is considered more fair.

Table 3: FID score for three 'super-categories': 'Head', 'Body', and 'Tail'.

| Dataset | FID↓ $P_{category}$ / Method | Head $\sim 80\%$ | Body $\sim 17\%$ | Tail $\sim 3\%$ | All |
|---|---|---|---|---|---|
| TinyImageNet-LT | DDPM (Ho et al., 2020) | 21.27 | 33.25 | 40.12 | 18.67 |
| | CBDM (Qin et al., 2023) | 24.12 | 35.02 | 48.07 | 20.90 |
| | OCLT (Zhang et al., 2024) | **20.64** | 30.62 | 39.67 | 17.72 |
| | **CCUA (ours)** | 21.32 | **27.98** | **30.39** | **15.24** |
| Places-LT | DDPM (Ho et al., 2020) | 19.26 | 20.31 | 23.74 | 13.89 |
| | CBDM (Qin et al., 2023) | 21.26 | 21.68 | 26.07 | 15.15 |
| | OCLT (Zhang et al., 2024) | 19.16 | 19.43 | 22.84 | 13.04 |
| | **CCUA (ours)** | **18.15** | **19.27** | **20.84** | **11.99** |

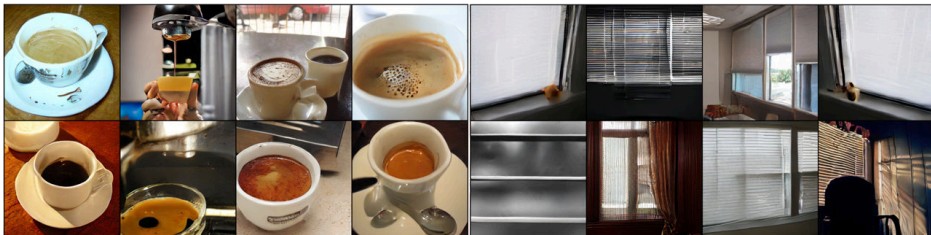

(a) Synthetic images from SiT (Ma et al., 2024).

(b) Synthetic images from CCUA (ours).

Figure 4: Synthetic images of SiT and it with CCUA for ImageNet-LT tail classes 'espresso' and 'window shade'. All methods start the denoising process from the same Gaussian noise at corresponding grid cells. CCUA shows consistently higher diversity and fidelity compared to SiT.

## 4.3 QUALITATIVE RESULTS AND ABLATION STUDY

**Visualization of Synthetic Images** Fig. 4 shows synthetic images of SiT and with CCUA for ImageNet-LT tail classes 'espresso' and 'window shade'. Compared to SiT, CCUA achieves visually higher fidelity and diversity. More qualitative results on ImageNet-LT, TinyImageNet-LT, and CIFAR100-LT are shown in appendix A.4.1 and A.4.3.

Table 4: Hyper-parameters Analysis of CCUA on ImageNet-LT $256 \times 256$ with DiT-based models. All models are trained from scratch to 40 epochs.

| Method | $\alpha$ | $\gamma$ | IS ↑ | FID ↓ | sFID ↓ | Prec. (%) ↑ | Recall (%) ↑ |
|---|---|---|---|---|---|---|---|
| SiT (baseline) | - | - | 53.93 | 33.87 | 22.66 | 54.56 | 19.17 |
| **CCUA (ours)** | 1.0 | 1.0 | 52.54 | 32.96 | **16.26** | 48.92 | **29.90** |
| | 0.5 | 0.5 | 62.26 | 30.07 | 20.32 | 55.55 | 22.29 |
| | 0.05 | 0.05 | **73.60** | **25.89** | 16.54 | 58.62 | 23.05 |
| | 0.01 | 0.01 | 65.79 | 28.61 | 18.97 | 58.52 | 19.75 |
| | 0.05 | 0 | 64.28 | 28.70 | 18.69 | 57.54 | 21.25 |
| | 0 | 0.05 | 64.65 | 29.04 | 20.17 | **58.64** | 19.97 |

Table 5: Batch Resample Strategy Analysis on ImageNet-LT. All models are trained from scratch to 40 epochs. Blue '()' highlights the improvement of each method compared to the SiT baseline, while Red '()' highlights the decline compared to SiT baseline.

| Method | IS ↑ | FID ↓ | sFID ↓ | Prec. (%) ↑ | Recall (%) ↑ |
|---|---|---|---|---|---|
| SiT (baseline) | 53.93 | 33.87 | 22.66 | 54.56 | 19.17 |
| +Batch Resample | 71.39 (+17.46) | 28.05 (-5.82) | 21.57 (-1.09) | 57.61 (+3.05) | 17.26 (-1.91) |
| $\mathcal{L}_{ccua}$ (Eq. 7) | 57.79 (+3.86) | 32.25 (-1.62) | 20.71 (-1.95) | 54.47 (-0.09) | 20.66 (+1.49) |
| +Batch Resample | **73.60** (+19.67) | **25.89** (-7.98) | **16.54** (-6.12) | **58.62** (+4.06) | **23.05** (+3.88) |

Table 6: Batch Resample Strategy Analysis on TinyImageNet-LT.

| Method | FID ↓ | $FID_{tail}$ ↓ | KID↓ |
|---|---|---|---|
| DDPM (baseline) | 18.67 | 40.12 | 6.31 |
| +Batch Resample | 17.18 (-1.49) | 33.60 (-6.52) | 4.66 (-1.65) |
| $\mathcal{L}_{ccua}$ (Eq. 7) | 17.16 (-1.51) | 37.88 (-2.24) | 4.89 (-1.42) |
| +Batch Resample | 15.24 (-3.43) | 30.39 (-9.73) | 3.83 (-2.48) |

**Ablation Study of Loss Function** Table 4 shows the ablation study of the hyper-parameters for the proposed $\mathcal{L}_{ucl}$ and $\mathcal{L}_{al}$ losses, which is conducted on ImageNet-LT dataset. We found that extremely large $\alpha = 1$ and $\gamma = 1$ would cause much higher Recall and lower Precision, which means the generated images possess better diversity but worse fidelity. We selected the optimal $\alpha = 0.05$, $\gamma = 0.05$ for the best FID while trading off diversity and fidelity.

**Ablation Study of Batch Resample Strategy** As shown in Table 5, we measure the performance of SiT and CCUA, with or without applying batch resampling strategy. Compared to SiT, the proposed CCUA loss achieves better IS, FID, sFID and Recall with only **0.09** Precision's decline, illustrates our method on improving diversity without trading off fidelity. With applying batch resample strategy, the CCUA further improves all metrics. Notes that the CCUA increases Recall from **20.66** to **23.05**, while SiT suffers a decline in Recall from **19.17** to **17.26** when applying batch resampling. As we discuss in Section. 3.1, the $\mathcal{L}_{ucl}$ could benefit from batch resample strategy due to the increasing instances of tail classes, leading to more diverse distribution in the latent space. We also report batch resample ablation study on TinyImageNet-LT in Table 6.

### 4.4 LIMITATION OF OUR METHOD

Our MSE loss involves both conditional generation and unconditional generation, which requires two passes of the denosing network during training and increases training time. In practice, with applying our method, the training time for one epoch is 1.6x of that of DDPM, and 1.48x of that of SiT. However, our method doesn't increase inference latency. With optimized implementation, the training time can be reduced to 1.3x of that of SiT, see Appendix A.2 for details.

### 5 CONCLUSION

Real-world data for training image generation models often exhibit long-tailed distributions. Similar to class-imbalanced GANs (Khorram et al., 2024), class-imbalanced diffusion models generates inferior tail class images due to data scarcity. We propose a framework with two losses. Firstly, we employ an unsupervised contrastive loss with negative samples only to contrast the latents of different synthetic images at every denoising step, promoting intra-class diversity in particular for tail classes. Our second loss aligns class-conditional generation with unconditional generation for large timesteps. This encourages the initial denoising steps to be class-agnostic, thereby enriching tail classes through knowledge sharing from head classes–a principle demonstrated to enhance long-tailed GAN performance (Shahbazi et al., 2022; Khorram et al., 2024), which we successfully adapt to diffusion models. With the two losses, our framework of contrastive conditional-unconditional alignment boosts the performane of DDPM (Ho et al., 2020) and SiT (Ma et al., 2024) for long-tailed image generation and outperforms alternative methods including CBDM (Qin et al., 2023), OCLT (Zhang et al., 2024), and Dispersive Loss (Wang & He, 2025). Extensive experiments on multiple datasets in particular ImageNet-LT with 256x256 resolution demonstrated the effectiveness of our method on both UNet-based architecture and Diffusion Transformer.

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

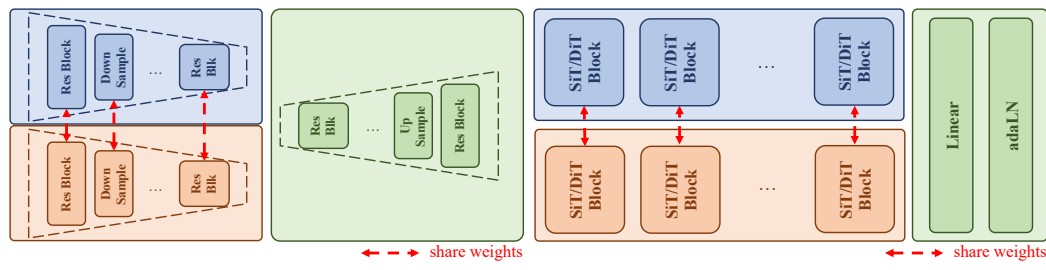

(a) w/ UNet-based Model          (b) w/ Diffusion Transformer

Figure 5: Model details of CCUA framework with UNet-based Model and Diffusion Transformer. As shown in Fig. 2, the noise estimation network is divided into two parts, a latent encoded network $e(*)$ and a decoded network $d(*)$. (a) For UNet-based architecture, $e(*)$ is defined as the UNet encoder, while $d(*)$ is defined as the UNet decoder. (b) For Diffusion Transformer, $e(*)$ is defined as all the SiT/DiT blocks, while $d(*)$ is defined as the final linear and adaLN projection layer.

Table 7: Our method outperforms other baselines on all datasets. We also provide the results of DDPM trained on balanced datasets, which show the upper bound of performance. All models are measured with DDIM (Song et al., 2020) 100 steps for conditional generation with CFG. Blue '()' shows improvement of our method over DDPM baseline (Ho et al., 2020). Green '()' shows improvement of DDPM trained on balanced version over trained on long-tailed version.

| Dataset | Method | FID↓ | FID$_{tail}$ ↓ | KID$_{\times 1k}$ ↓ |
|---|---|---|---|---|
| CIFAR10-LT | DDPM$^*_{bal}$ (Ho et al., 2020) | 4.90 (-1.03) | 6.27 (-5.98) | 1.32 (-0.32) |
| | DDPM (Ho et al., 2020) | 5.93 | 12.25 | 1.64 |
| | CBDM (Qin et al., 2023) | 5.81 | **10.01** | 1.58 |
| | OCLT (Zhang et al., 2024) | 6.10 | 11.13 | 1.58 |
| | **CCUA (ours)** | **5.56** (-0.37) | 10.03 (-2.22) | **1.27** (-0.37) |
| CIFAR100-LT | DDPM$^*_{bal}$ (Ho et al., 2020) | 5.15 (-1.80) | 8.97 (-8.48) | 1.05 (-0.66) |
| | DDPM (Ho et al., 2020) | 6.95 | 17.45 | 1.71 |
| | CBDM (Qin et al., 2023) | 6.50 | 17.36 | 1.41 |
| | OCLT (Zhang et al., 2024) | 6.45 | 17.22 | 1.42 |
| | **CCUA (ours)** | **6.24** (-0.71) | **16.35** (-1.10) | **1.36** (-0.35) |

# A   TECHNICAL APPENDICES AND SUPPLEMENTARY MATERIAL

## A.1   IMPLEMENTATION DETAILS

**Dataset Details**    We keep the original $32 \times 32$ resolution for CIFAR10-LT/CIFAR100-LT, and resize images to $64 \times 64$ for TinyImageNet-LT and Places-LT, while $256 \times 256$ for ImageNet-LT. Same as (Qin et al., 2023; Yan et al., 2024; Zhang et al., 2024), we adopt the same methodology presented in (Cao et al., 2019) to construct long-tailed version datasets with an imbalance factor of $0.01$.

**Model Architecture Details**    As described in Sec. 3, the proposed CCUA framework can be applied into UNet-based architecture and Diffusion Transformer. In Fig. 5, we display our UNet-based model and Diffusion Transformer in details. As shown in Fig. 2, the noise estimation network is divided into two parts, a latent encoded network $e(*)$ and a decoded network $d(*)$. In our setting, for UNet-based model, $e(*)$ is defined as the UNet encoder, while $d(*)$ is defined as the UNet decoder, as shown in Fig. 5 (a). For Diffusion Transformer, $e(*)$ is defined as all the SiT/DiT blocks, while $d(*)$ is defined as the final linear and adaLN projection layer, as shown in Fig. 5 (b).

**Training Details**    For SiT pipeline, we strictly follow the same training hyper-parameter settings as Dispersive Loss (Wang & He, 2025). We use 4 A6000 GPUs to train all SiT based models on ImageNet-LT datasets with batch size $48$. During the training process, all methods are trained from scratch. We report 40 epochs, 80 epochs, 120 epochs, and 160 epochs results in Table 1. For DDPM pipeline, we follow the same training configurations of the baseline models (Ho et al., 2020; Qin et al., 2023; Zhang et al., 2024). We use one RTX 4090 GPU to train each model on CIFAR10-

LT/CIFAR100-LT datasets with batch size $64$ while using 2 A100 GPUs for TinyImageNet-LT and Places-LT, with batch size $128$. During the training process, all methods are trained from scratch for 200k iterations on CIFAR10-LT/CIFAR100-LT datasets, 100k iterations on TinyImageNet-LT and Places-LT datasets, while they follow the classifier-free guidance (CFG) algorithm (Ho & Salimans, 2022), which randomly drops labels with a probability of 10%. On ImageNet-LT and TinyImageNet-LT datasets, we apply batch resample strategy with the re-balanced factor $0.1$, while we don't apply batch resample strategy on Places-LT and CIFAR10-LT/CIFAR100-LT datasets. For all datasets, we also apply timestep adaptive weight $t/T$ to unsupervised contrastive loss since we find the unsupervised contrastive loss could also benefit from such an adaptive weight, like alignment loss.

**Evaluation Details**  In Table 2, $FID_{tail}$ denotes the FID score for synthetic images of the last 30% classes in each dataset. Specifically, we categorize the 'Tail' classes as the last 66 classes for TinyImageLT (200 classes), the last 121 classes for Places-LT (365 classes), respectively. In Table 1, the evaluation metrics are based on 50k synthetic images generated by each method with a CFG strength 7.5. In Table 2, the evaluation metrics are based on 10k synthetic images generated by each method. During inference, we perform a grid search algorithm for each method to determine the optimal guidance strength $\omega$ of CFG, ensuring each model achieves its best performance.

## A.2  ACCELERATED IMPLEMENTATION

CCUA incurs longer training time for diffusion models. We further optimized our implementation by a simple trick of computing conditional generation and unconditional generation for the same batch with one function call of model.forward(). In our original implementation, we called model.forward() twice in each iteration, which is not as efficient.

```
# x_batch.shape: B, C, H, W
# original implementation for one iteration
cond_output = model.forward(x_batch, c)
uncond_output = model.forward(x_batch, null)

# optimized implementation for one iteration
cond_output, uncond_output = model.forward(
    torch.cat([x_batch, x_batch], dim=0),
    torch.cat([c, null], dim=0)
).chunk(2, dim=0)
```

At the cost of increased GPU memory consumption, our optimized implementation is only 1.3x slower than SiT with original diffusion loss.

| Method | Average Training Time | GPU Consumption |
|---|---|---|
| SiT | 8.7 steps/s | 0.44 GB/image |
| CCUA (original implementation) | 5.8 steps/s | 0.48 GB/image |
| CCUA (accelerated implementation) | 6.7 steps/s | 0.52 GB/image |

## A.3  MORE QUANTITATIVE RESULTS

**Extremely Imbalanced Generation on ImageNet-LT dataset**  We measure the performance of CCUA and SiT w.r.t the imbalanced factor 0.001 for ImageNet-LT dataset. Note that for ImageNet datasets, each class only contains 1300 images, which means with 0.001 imbalanced factor the tail classes only contains 1 2 images. Our method improves the baseline for such a challenging dataset.

**Class-imbalanced Generation on CIFAR10-LT/CIFAR100-LT datasets**  We conduct more experiments and analysis on CIFAR10-LT/CIFAR100-LT datasets, as shown in Table 7 and Table 9. We provide the metrics of the DDPM model trained on the balanced version, i.e., the original CIFAR10/CIFAR100 datasets, denoted by $DDPM^*_{bal}$, as the theoretical optimal reference. On CIFAR10-LT/CIFAR100-LT, our method achieves the lowest FID and KID compared to baseline methods. Note that the FID gap between DDPM and $DDPM^*_{bal}$ is **1.03** on CIFAR10-LT and **1.8** on CIFAR100-LT, while our method improves FID **0.37** over 1.03 on CIFAR10-LT and **0.71** over 1.8 on

Table 8: Comparison on ImageNet-LT $256 \times 256$ with SiT pipeline with imbalanced factor 0.001.

| Epochs | Method | IS ↑ | FID ↓ | sFID ↓ | Prec. (%) ↑ | Recall (%) ↑ |
|--------|--------|------|-------|--------|-------------|--------------|
| 40 | SiT | 29.89 | 51.26 | 21.26 | 38.09 | 22.04 |
| | +CCUA (ours) | **48.79** | **36.41** (-14.85) | **16.41** | **45.04** | 21.26 |
| 80 | SiT | 46.47 | 37.55 | 23.50 | 48.66 | 20.39 |
| | +CCUA (ours) | **83.93** | **25.96** (-11.59) | **17.44** | **55.91** | 19.33 |
| 120 | SiT | 59.05 | 30.56 | 22.05 | 54.79 | 20.61 |
| | +CCUA (ours) | **105.86** | **22.44** (-8.12) | **18.40** | **60.34** | 19.29 |
| 160 | SiT | 67.70 | 27.25 | 19.96 | 57.49 | 18.89 |
| | +CCUA (ours) | **118.16** | **20.93** (-6.32) | **18.88** | **62.88** | 18.89 |

Table 9: We compare our method with other baselines on CIFAR10-LT/CIFAR100-LT datasets, with DDPM 1000 steps for conditional generation with CFG. We also report the data augmentation method ADA (Karras et al., 2020) and $\omega$-scheduler (Wang et al., 2024), which are orthogonal to our method.

| Method | CIFAR10-LT | | CIFAR100-LT | |
|--------|------------|------|-------------|------|
| | FID↓ | IS↑ | FID↓ | IS↑ |
| DDPM$^*_{bal}$ (Ho et al., 2020) | 4.87 | 9.35 | 5.20 | 13.29 |
| DDPM (Ho et al., 2020) | 5.81 | 9.36 | 7.09 | 12.64 |
| +ADA (Karras et al., 2020) | - | - | 6.69 | 12.87 |
| +$\omega$-Scheduler (Wang et al., 2024) | 5.87 | 9.22 | 6.60 | 12.10 |
| CBDM (Qin et al., 2023) | 5.92 | 9.38 | 6.52 | 12.79 |
| OCLT (Zhang et al., 2024) | 5.69 | 9.42 | 6.23 | **13.18** |
| **CCUA (ours)** | **5.57** | **9.42** | **5.99** | 13.01 |

CIFAR100-LT, achieving $> \mathbf{35\%}$ performance improvement. To investigate the consistency of such improvements, we compare our method and all baseline methods on CIFAR10-LT and CIFAR100-LT with DDPM 1000 steps. As shown in Tab. 9, our method achieves consistent improvements of FID for full 1000 sampling steps. We also compare to a widely used data augmentation technique, Adaptive Discriminator Adaption (ADA) (Karras et al., 2020) for generative models on the full DDPM 1000 sampling steps. Besides, we apply the CFG guidance strength scheduler $\omega$-cos (Wang et al., 2024) on DDPM, which gradually increases the guidance strength during sampling time steps decreasing to force the model transfer from unconditional to conditional generation.

**Consistent Improvement for Fewer Sampling Steps** To investigate the model's performance on extremely few sampling steps, We evaluate our method and all baselines for DDIM 10 steps with CFG conditional generation on CIFAR10-LT/CIFAR100-LT. As shown in Tab. 10, our method achieves the best FID, $\text{FID}_{tail}$ and KID scores. Our method achieves even better results than the theoretical optimal

Table 10: We compare our method with other baselines on CIFAR10-LT/CIFAR100-LT datasets with DDIM 10 sampling steps for conditional generation with CFG. Blue '()' shows improvement of our method over DDPM baseline (Ho et al., 2020). Green '()' shows improvement of DDPM trained on balanced version over trained on long-tailed version.

| Dataset | Method | FID↓ | $\text{FID}_{tail}$ ↓ | $\text{KID}_{\times 1k}$ ↓ |
|---------|--------|------|------------------------|-----------------------------|
| CIFAR10-LT | DDPM$^*_{bal}$ | 13.28 (-1.44) | 13.26 (-5.01) | 6.06 (-0.98) |
| | DDPM | 14.72 | 18.27 | 7.04 |
| | CBDM | 13.54 | 16.90 | 6.52 |
| | OCLT | 15.48 | 20.73 | 7.39 |
| | **CCUA (ours)** | **13.16** (-1.56) | **16.83** (-1.44) | **6.04** (-1.00) |
| CIFAR100-LT | DDPM$^*_{bal}$ | 13.34 (-0.75) | 18.27 (-6.80) | 5.56 (-0.57) |
| | DDPM | 14.09 | 25.07 | 6.13 |
| | CBDM | 13.37 | 23.97 | 5.83 |
| | OCLT | 13.70 | 24.48 | 5.73 |
| | **CCUA (ours)** | **12.90** (-1.19) | **23.17** (-1.90) | **5.63** (-0.50) |

Table 12: Ablation Study of Latent Encoder for DiT-based model.

| Epochs | Latent | IS ↑ | FID ↓ | sFID ↓ | Prec. (%) ↑ | Recall (%) ↑ |
|--------|--------|------|-------|--------|-------------|--------------|
| 40 | **N/4** | 70.69 | 27.04 | 20.50 | **60.16** | 20.12 |
| | **N/2** | 68.57 | 28.00 | 21.13 | 58.59 | 21.28 |
| | **3N/4** | 68.48 | 27.99 | 19.96 | 58.96 | 21.09 |
| | **N** | **73.60** | **25.89** | **16.54** | 58.62 | **23.05** |
| 80 | **N/4** | **111.91** | **19.43** | 18.25 | **69.44** | 18.88 |
| | **N/2** | 105.41 | 20.10 | 19.49 | 68.22 | 19.24 |
| | **3N/4** | 109.29 | 19.78 | 18.50 | 69.30 | 18.80 |
| | **N** | 103.66 | 19.99 | **14.88** | 65.80 | **21.31** |
| 120 | **N/4** | **140.46** | **16.29** | 17.19 | **73.92** | 18.40 |
| | **N/2** | 134.89 | 16.63 | 17.89 | 72.94 | 18.86 |
| | **3N/4** | 137.21 | 16.65 | 17.46 | 73.60 | 18.41 |
| | **N** | 119.11 | 17.55 | **13.89** | 68.38 | **21.20** |
| 160 | **N/4** | **153.07** | **15.08** | 16.54 | **75.73** | 17.27 |
| | **N/2** | 148.61 | 15.14 | 16.94 | 75.40 | 18.11 |
| | **3N/4** | 148.88 | 15.22 | 16.80 | 75.32 | 19.35 |
| | **N** | 124.87 | 16.41 | **13.17** | 69.84 | **21.34** |

model $DDPM^*_{bal}$ under such an extreme experimental setting. Such an improvement demonstrates the effectiveness of our method for training long-tailed image generation diffusion model.

**Class-imbalanced Unconditional Generation** We also evaluate all the models for class-imbalanced unconditional generation. As shown in Tab. 11, the proposed method reduces FID from **27.52** to **24.16** for CIFAR10-LT and from **18.53** to **15.97** for CIFAR100-LT. Such an improvement in unconditional generation highlights the effectiveness of the proposed contrastive learning loss, particularly the unsupervised contrastive loss $\mathcal{L}_{ucl}$.

Table 11: Unconditional generation w/ DDIM 100 steps.

| Method | CIFAR10-LT | | CIFAR100-LT | |
|--------|-----------|-----|------------|-----|
| | FID↓ | IS↑ | FID↓ | IS↑ |
| DDPM | 27.52 | 6.65 | 18.53 | 8.68 |
| CBDM | 25.60 | 6.70 | 17.06 | 9.00 |
| OCLT | 31.38 | 6.34 | 18.97 | 8.73 |
| **Ours** | **24.16** | **6.80** | **15.97** | **9.23** |

**Discussion of Latent Encoder of SiT** We conduct the ablation study to investigate the choice of latent encode network in SiT. For a SiT model with $N$ SiT/DiT blocks, we use the latent representations $h$ from the $N/4$-th, $N/2$-th, $3N/4$-th, and the $N$-th block for $\mathcal{L}_{ucl}$ loss. As shown in Table 12, we found even the last layer does not always provide the best performance on all metrics, it has the best spatial FID (sFID) score and Recall. These two metrics reflect the better quality and diversity of representations in latent space, thus we select the N-th layer, i.e. the last SiT block, as the encoder.

**Statistical Significance Analysis** To inspect the robustness of the proposed method, we conduct statistical significance analysis on the ddpm and our method, as shown in Tab. 13. With applying different random seed, the FID deviation of DDPM is about 0.04 while our method is about 0.01, illustrating the robustness of our method for different random seeds.

Table 13: Statistical Analysis for random seed on CIFAR100-LT, sampling with DDIM 100 steps.

| Seed | Method | FID↓ | Method | FID↓ |
|------|--------|------|--------|------|
| 0 | DDPM | 7.02 | Ours | 6.26 |
| 42 | DDPM | 6.95 | Ours | 6.24 |
| 2025 | DDPM | 7.04 | Ours | 6.27 |

## A.4 MORE QUALITATIVE RESULTS

### A.4.1 MORE VISUALIZATION OF SYNTHETIC IMAGES ON IMAGENET-LT AND TINYIMAGENET-LT

Fig. 6 shows synthetic images of SiT and with the proposed CCUA for ImageNet-LT tail classes 'bubble', 'redwine', 'comic book' and 'yawl'. Fig. 7 shows synthetic images of DDPM and with the proposed CCUA for TinyImageNet-LT tail classes 'teapot', 'water tower', 'pretzel', 'mushroom', 'orange', and 'pizza'. These methods start the denoising process from the same Gaussian noise at corresponding grid cells. As shown in Fig. 6 and Fig. 7, synthetic images of CCUA show consistently higher diversity and fidelity compared to SiT.

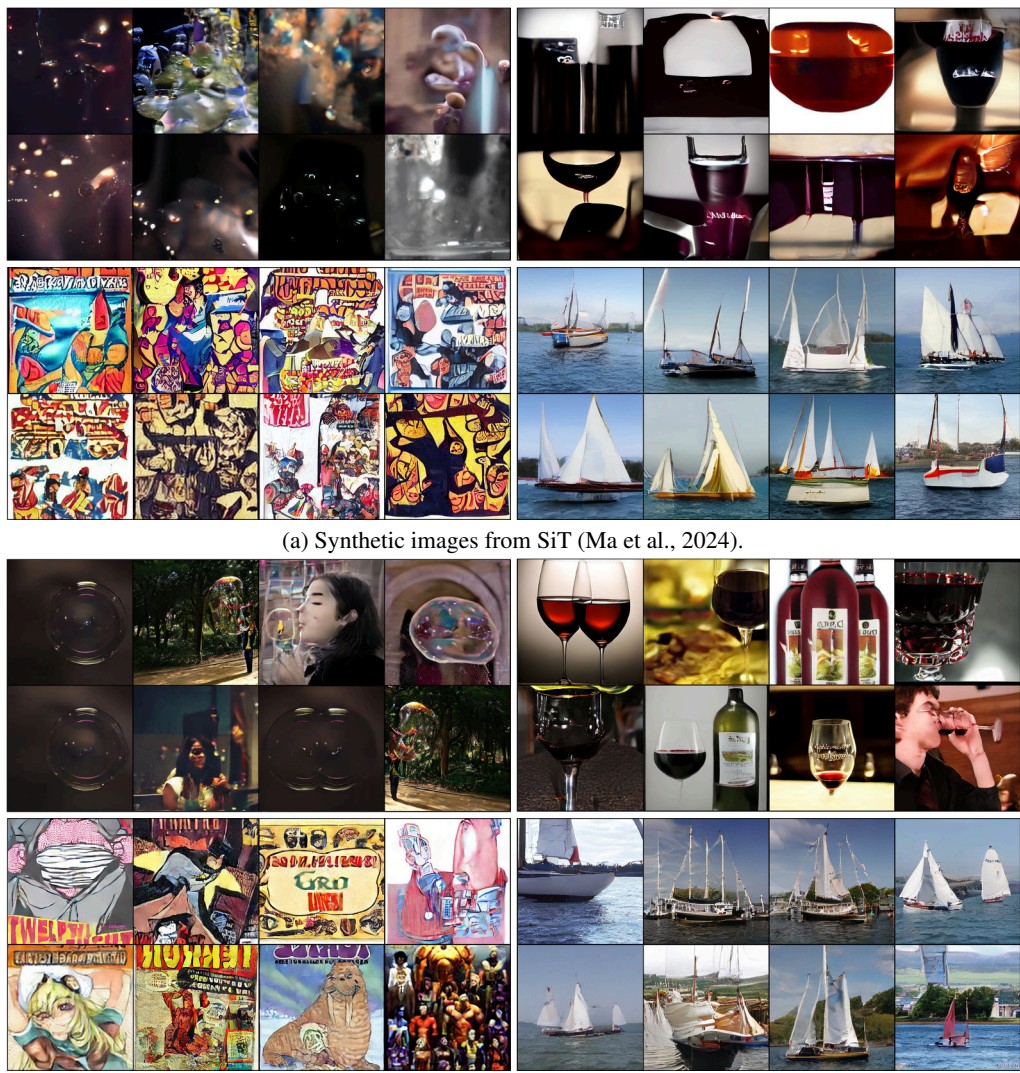

(a) Synthetic images from SiT (Ma et al., 2024).

(b) Synthetic images from CCUA (ours).

Figure 6: Synthetic images of SiT and CCUA for ImageNet-LT tail classes (from top-left to right-bottom: 'bubble', 'redwine', 'comic book' and 'yawl'). All methods start the denoising process from the same Gaussian noise at corresponding grid cells. CCUA shows consistently higher diversity and fidelity compared to SiT.

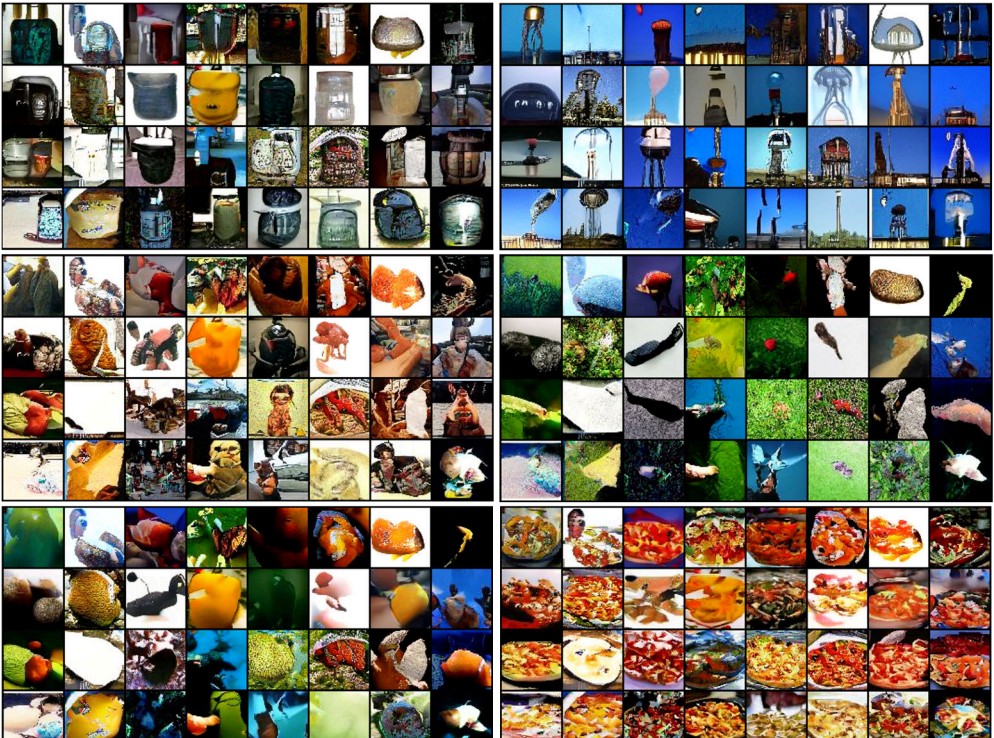

(a) Synthetic images from DDPM (Ho et al., 2020).

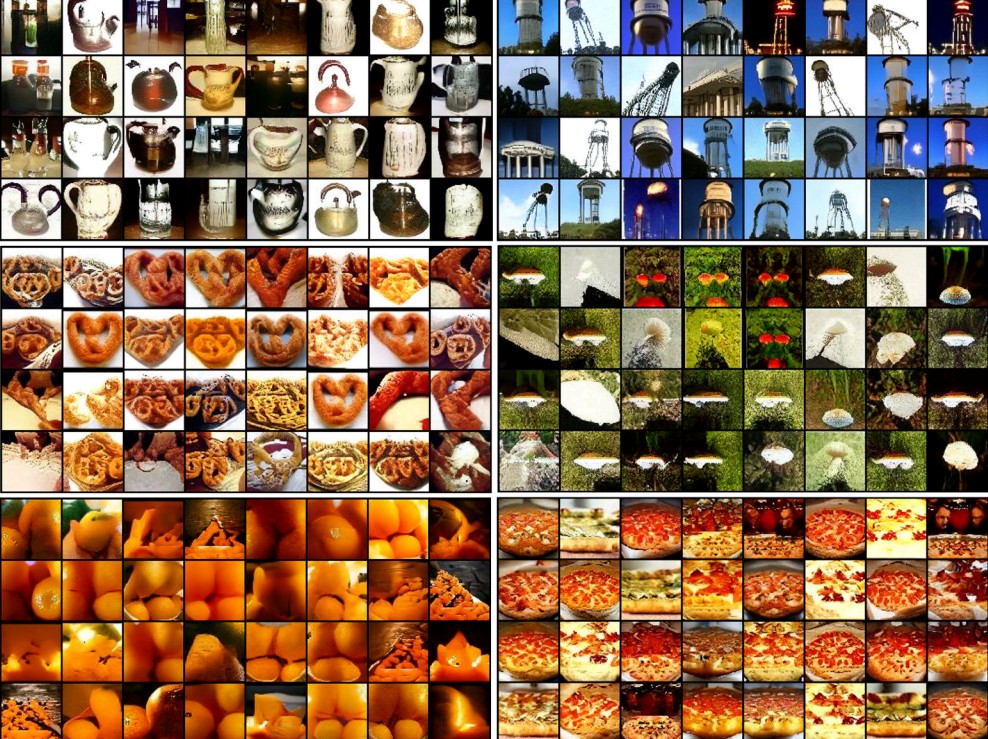

(b) Synthetic images from CCUA (ours).

Figure 7: More synthetic results for TinyImageNet-LT tail classes (from top-left to right-bottom: 'teapot', 'water tower', 'pretzel', 'mushroom', 'orange', and 'pizza'). Images in corresponding grid cell for DDPM and our method are initialized from the same Gaussian noise. Our method achieves more diverse images with higher fidelity for tail classes. Note that for 'pretzel' and 'orange' classes, DDPM fails to synthesize images correlated to the class while CCUA synthesizes diverse images with high quality.

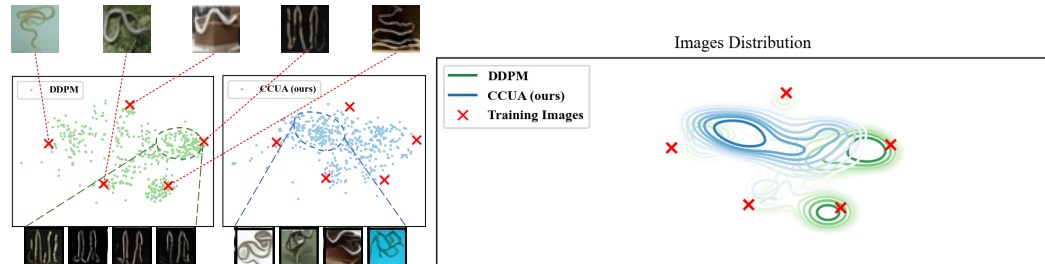

(a) Images in low-dimensional embeddings.      (b) Density distribution of synthetic images.

Figure 8: (a) Visualization of low-dimensional embeddings of five training images x for a tail class ('worm' in CIFAR100-LT) and synthetic images generated by DDPM (Ho et al., 2020) and our method. (b) We also show the distributions of real images and synthetic images generated by our method and the original DDPM.

### A.4.2   DISTRIBUTION OF SYNTHETIC IMAGES FOR TAIL CLASS

We visualize synthetic images in the feature space and their density function for our method and the original DDPM. Specifically, we use the Inception-V3 (Szegedy et al., 2016) model to extract 2048 dimensional features of each image, and then apply t-SNE (Van der Maaten & Hinton, 2008) to project these features into 2 dimensions. As shown in Fig. 8, the original DDPM overfits and generates highly similar images, while synthetic images based on our method have more diversity. We visualize the image distribution more specifically by using kernel density estimation in the bottom. The grey region represents the distribution of real 'worm' images from the class-balanced CIFAR100 dataset, while red 'x' points denote all 5 'worm' images from the class-imbalanced CIFAR100-LT dataset. The blue region represents the distribution of images synthesized by our method, while the green region is for the original DDPM. Areas with higher color saturation (dark, green, or blue) indicate regions of higher density, which correspond to modes of distribution. Synthetic images from DDPM mostly concentrates around the training images, leading to mode collapse. Synthetic images from our method spans the space enclosed by all training images, the distribution of which is shown in blue in (b).

### A.4.3   MODE COLLAPSE ISSUE ON TAIL CLASS

Fig. 9 shows synthetic images of baseline methods and our method for CIFAR100-LT tail classes 'rose' and 'table'. All methods start the denoising process from the same Gaussian noise at corresponding grid cells. Red dashed ellipses highlight the mode collapse issues observed in DDPM. With the same initial noise, our method gives synthetic images with higher diversity and fidelity compared to DDPM. For example, DDPM always generates rose images containing only one rose. Our method can generate an image containing two roses (see the sixth column of the last row). To further clarify, in the 'table' class, DDPM repeatedly produces near-identical images as highlighted in red ellipses. In contrast, our method produces a more varied set of table images. This suggests that our method can better capture greater diversity in image generation when compared to DDPM (Ho et al., 2020). To further illustrate how the proposed method mitigates the issue of overfitting, we visualize the top-10 nearest neighbors among 1000 synthetic images to an anchor image in the training set for the original DDPM and our method. As shown in Fig. 10, DDPM shows the mode collapse to the training image while our method's generated images show much better diversity. For example, in the first two rows, DDPM generates repeated vertical worms while our method generates worms with diverse directions. In the 9th-10th rows, DDPM generates tables with repeated modes while our method generates tables with different styles and colors.

### A.4.4   MORE VISUALIZATION OF REVERSE PROCESS OF DDPM FOR DIFFERENT CLASSES

To illustrate our observation in Section 3.2 more clearly, we decompose $x_t$ into a combination of low-frequency images and high-frequency images, as shown in Fig 11. The low-frequency images with the same initial noise are very similar for different classes for the initial steps, which is also observed in prior work (Si et al., 2024).

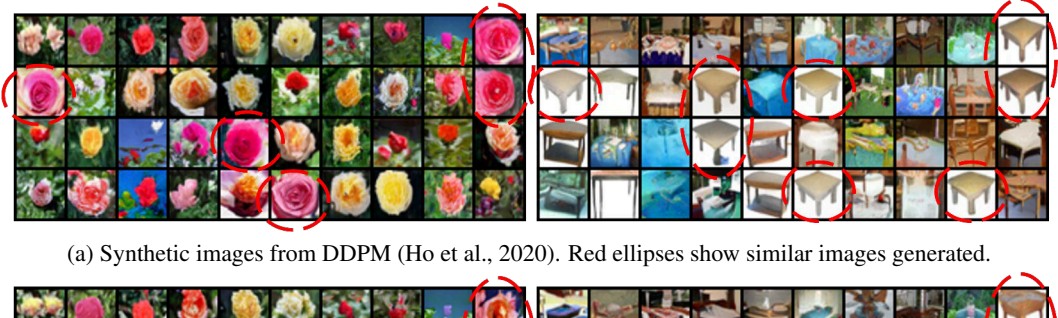

(a) Synthetic images from DDPM (Ho et al., 2020). Red ellipses show similar images generated.

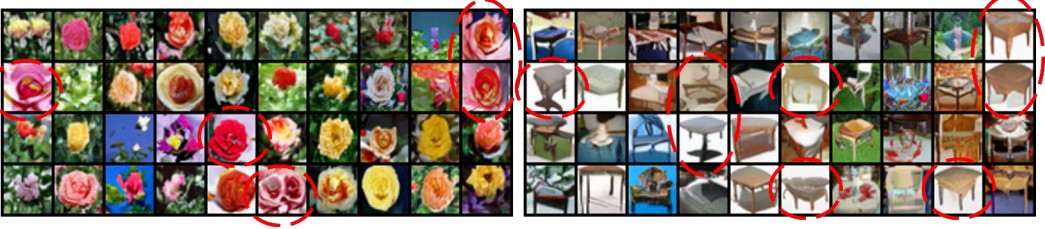

(b) Synthetic images from CCUA (ours) have more diversity and are less repetitive.

Figure 9: The synthesized results for CIFAR100-LT tail classes 'rose' and 'table' are shown for our method and baseline methods. All methods initiate reverse processing from the same Gaussian noise for images at corresponding grid cells. Red dashed ellipses highlight mode collapse issues observed in DDPM. Overall, our method demonstrates higher diversity and fidelity compared to DDPM baseline.

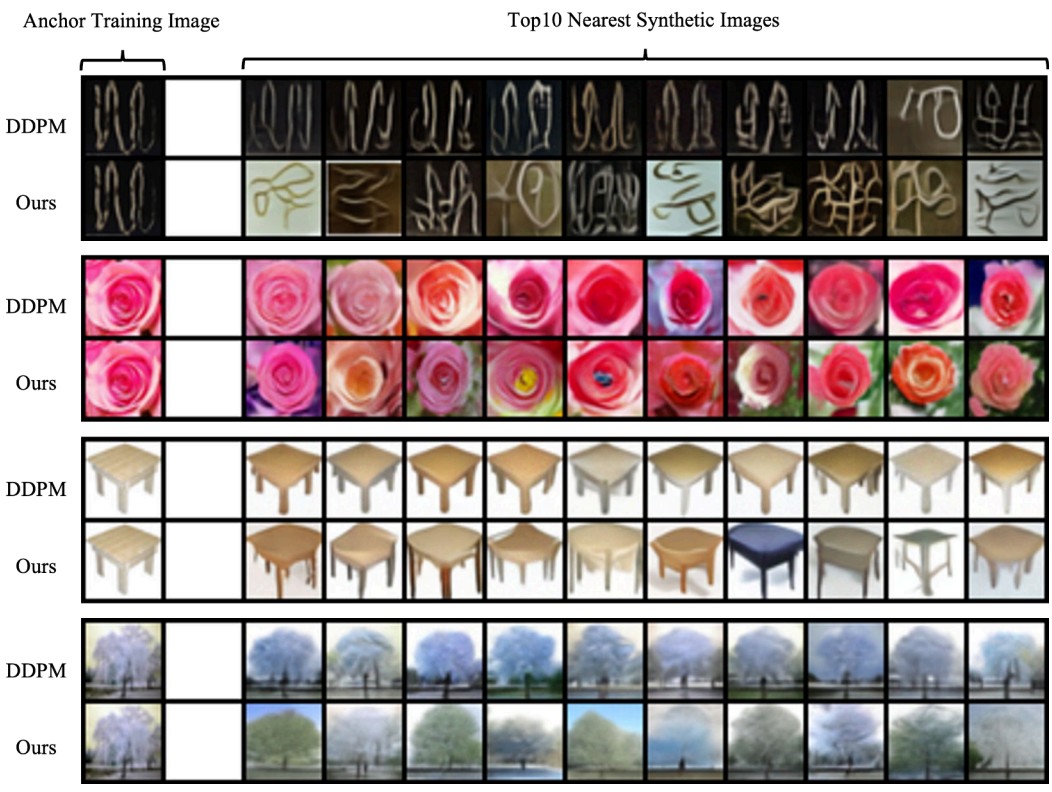

Figure 10: To see overfitting on tail classes, we find Top-10 nearest neighbors (3rd to 12th columns sorted by distances) among 1000 synthetic images to an anchor image (Leftmost column) in the training set. KNN is based on $L_2$ distances of Inception V3 embeddings. For each example, the top row is the results of DDPM, and the bottom row shows ours. The nearest neighbors from DDPM show the mode collapse to the training image, while our method generated more diverse images.

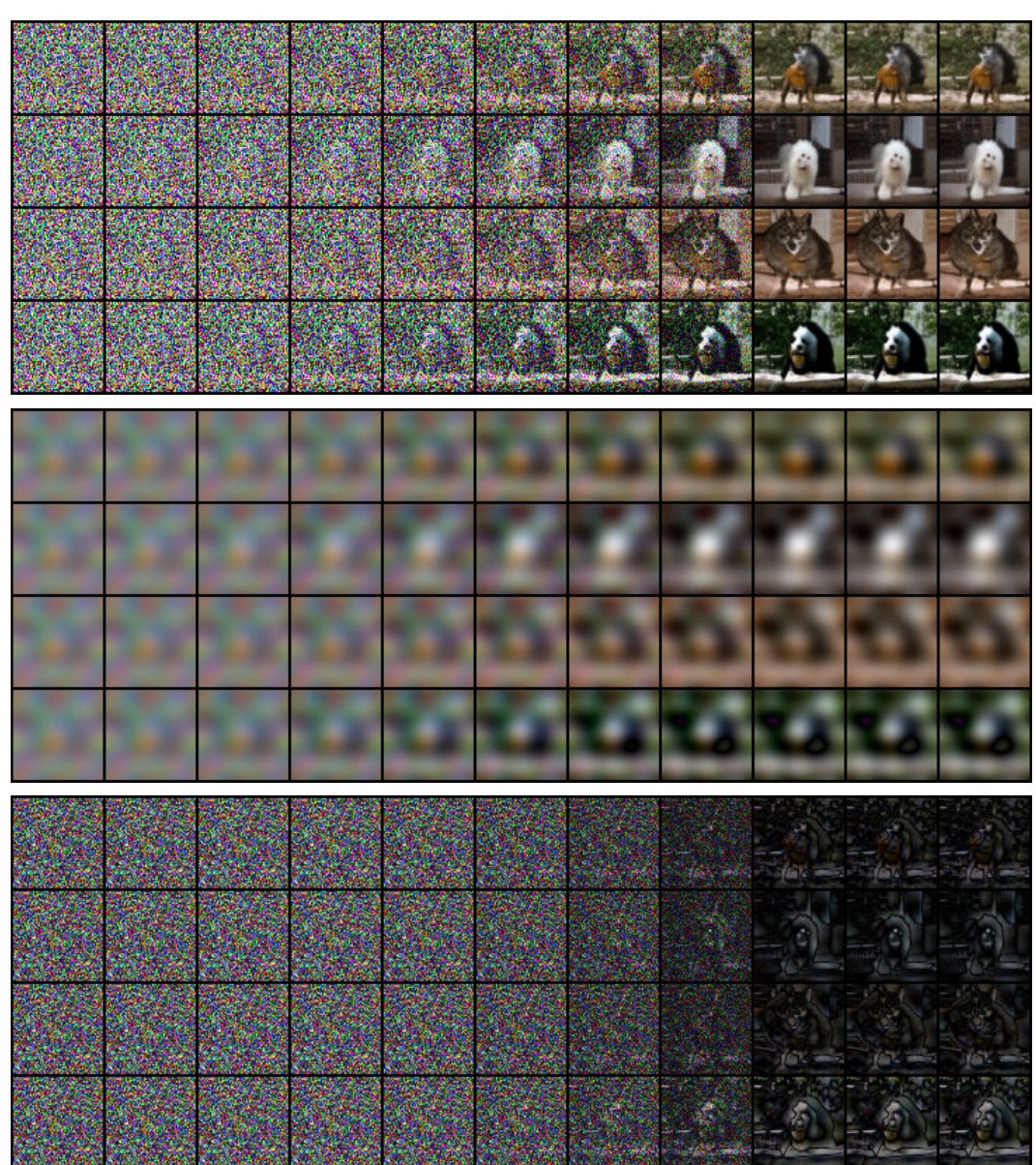

Figure 11: **Top:** reverse process starting from the same initial Gaussian noise but with different class conditions (2nd-4th rows) or without condition (1st row). **Middle:** low-frequency components of each noisy image. **Bottom:** high-frequency components of each noisy image.

## A.5 The Necessity of Combining Alignment Loss and Unconditional Contrastive Loss

Combining Alignment Loss and Unconditional Contrastive Loss on unconditional latents implicitly diversifies conditonal latents. Here, we give a theoretical explanation why this way is better than directly diversifying conditional latents via InfoNCE as is done in concurrent work of dispersive loss (Wang & He, 2025).

**Limitation of diversifying conditional latents directly**    It is known that InfoNCE loss (Oord et al., 2018) for contrastive learning gives a lower bound of mutual information between data sample and representation/latents. In other words, minimizing InfoNCE loss maximizes mutual information. We denote conditional latent as $h_t^c := e_\theta(x_t, t, c)$ and unconditional latent as $h_t^u := e_\theta(x_t, t, \emptyset)$, where $c$ is the class condition. The InfoNCE loss on unconditional latents maximizes the mutual information $I(h_t^u; x)$ between image $x$ and its unconditional latent $h_t^u$. Similarly, InfoNCE loss on conditional latents  (Wang & He, 2025) maximizes the mutual information $I(h_t^c; x, c)$ between image $x$ and its conditional latent $h_t^c$ and condition $c$.

Based on the chain rule of mutual information, $I(h_t^c; x, c)$ can be decomposed into two pars:

$$I(h_t^c; x, c) = I(h_t^c; c) + I(h_t^c; x|c). \tag{8}$$

The first item $I(h_t^c; c)$ measures the mutual information between class $c$ and latent $h_t^c$, while the second item $I(x; h_t^c|c)$ measures the conditional mutual information. Intuitively, the first term tells how much class condition reveals about the latent, which reflects **inter-class diversity**. The second term measures conditional mutual information with condition $c$ which reflects **intra-class diversity**. To address the mode collapse issue for tail classes and increase diversity, it is apparent that we need to focus on maximizing the second item, i.e., conditional mutual information $I(h_t^c; x|c)$. However, it is likely that optimization is dominated by inter-class mutual information $I(h_t^c; c)$ which admits a trivial solution of having an identical latent for all images of the same class leading to maximum mutual information.

**Unconditional Contrastive Loss and Combination with Alignment Loss**    We choose to diversify unconditional latents through InfoNCE loss with negative samples only, which maximizes $I(h_t; x)$. In this case, there is no shortcut solution, and the model is forced to diversify all latents regardless of class. We further distill diversified unconditional latent to conditional latent with a simple alignment loss.

