# OpenReview forum: "Contrastive Conditional–Unconditional Alignment for Long-tailed Diffusion Model"
_ICLR.cc/2026/Conference — Submitted to ICLR 2026_

### Official Review · Reviewer_jyaR · 2025-10-26

**Soundness:** 3
**Presentation:** 3
**Contribution:** 2
**Rating:** 4
**Confidence:** 4

**Summary:**

This paper proposes a new framework named CCUA (Contrastive Conditional–Unconditional Alignment) to address the challenges of mode collapse and insufficient diversity or fidelity in conditional diffusion models (e.g., DDPM, SBDM) trained on long-tailed datasets. CCUA introduces two key loss components: (1) an Unsupervised Contrastive Loss (UCL) that operates in the unconditional latent space using only negative samples to enhance intra-class diversity by increasing the distance among generated samples; and (2) an Alignment Loss (AL) that aligns conditional and unconditional noise estimations during early denoising steps, facilitating knowledge transfer from head classes to tail classes. The proposed method is compatible with both U-Net and Transformer-based diffusion architectures, and demonstrates superior performance on multiple long-tailed benchmarks, including ImageNet-LT, Tiny ImageNet-LT, and Places-LT, particularly in improving diversity (Recall) and fidelity (FID) for tail classes.

**Strengths:**

1. The paper introduces an innovative approach to applying contrastive learning in diffusion models by using only negative samples, rather than both positive and negative pairs. Moreover, it creatively adapts the conditional–unconditional alignment concept from GANs to diffusion models, enabling knowledge sharing during early denoising stages — a notable conceptual transfer.

2. The proposed method achieves significant quantitative improvements across multiple long-tailed datasets, particularly in Recall (diversity) metrics, demonstrating its effectiveness in alleviating mode collapse and enhancing sample diversity.

3. The CCUA framework is clearly structured with two distinct loss components and is easy to integrate into existing DDPM/SBDM training pipelines. It is compatible with both U-Net and DiT architectures, and the paper provides a clear explanation of how UCL mitigates mode collapse.

4. The work provides a general and effective regularization strategy for addressing class imbalance in conditional diffusion models. It represents a meaningful advancement in improving diffusion models for real-world, long-tailed data distributions.

**Weaknesses:**

1. The paper claims to improve the generation quality of tail classes while maintaining the fidelity of head classes. However, key quantitative tables (e.g., Table 1) only report overall FID, IS, Precision/Recall, and $FID_{tail}$. To fully substantiate this claim, it would be important to include metrics directly reflecting the quality of head-class generations, such as $FID_{head}$.

2. In Section 2, the paper states that “our unsupervised contrastive loss is formulated in the unconditional latent space and is implicitly extended to conditional training through our conditional-unconditional alignment loss.” This statement is somewhat unclear, as Equation (7) in Section 3.3 defines $\mathcal{L}\_{ucl}$ and $\mathcal{L}\_{al}$ as two independent weighted losses. The mechanism by which $\mathcal{L}\_{al}$​ implicitly extends or interacts with $\mathcal{L}\_{ucl}$​ is not sufficiently explained.

3. As mentioned in Section 2, _Dispersive Loss_ was originally designed for balanced datasets rather than long-tailed ones. A stronger and more convincing comparison would adapt _Dispersive Loss_ to the long-tailed setting to better isolate the specific contribution of the proposed alignment loss $\mathcal{L}\_{al}$​.

4. To apply $\mathcal{L}\_{ucl}$, the paper divides the noise estimation network $\epsilon\_{\theta}$​ into an encoder and a decoder. While this separation is natural for the U-Net architecture, the definition becomes less clear for the Transformer-based SiT, where all SiT/DiT blocks are treated as the encoder and only the final linear layer as the decoder. The rationale for treating all blocks’ outputs as the contrastive latent space, rather than using a bottleneck or intermediate representation, is not well justified or empirically validated.

**Questions:**

1. The paper computes the UCL in the unconditional latent space $e(\*;\emptyset)$ and transfers its effect to the conditional model via $\mathcal{L}\_{al}$​. Could the authors clarify why UCL is not directly applied in the conditional latent space $e(\*;c)$? A theoretical justification or comparison experiment would help explain whether applying UCL directly in the conditional space would indeed degrade performance.

3. In the experimental setup (Section 4.1), the compared methods for U-Net architectures using contrastive learning or explicit diversity enhancement appear to be limited to works published in or before 2024. Given the rapid progress in this area, it would be helpful to discuss or compare with more recent (2025 or latest) SOTA diffusion models that share similar motivations.

4. The proposed UCL computes the contrastive objective using only negative samples, aiming to enhance diversity by maximizing inter-sample distances. However, standard contrastive learning typically includes positive-pair attraction terms. It would be useful to provide further justification or an ablation study to clarify whether including positive pairs (same-class samples) would negatively affect long-tailed generation, and whether relying solely on repulsive forces might lead to overly dispersed latent representations (i.e., excessive intra-class variance).

---

> ### Author Response · Authors · 2025-11-23
> **Response 1/2 to Reviewer jyaR**
>
> We thank reviewer jyaR for the critical review. We appreciate the acknowlegement that our approach is innovative, general, and effective, and our framework clearly-structured with significant quantitative improvement. We provide responses to questions and raised weaknesses as follows.
>
> **Q1: include metrics directly reflecting the quality of head-class generations, such as FID$_{head}$.**
>
> A1: We include FID_head and FID_tail for DiT-based methods on ImageNet-LT as shown in Table 1 in our revised submission. Compared to SiT, CCUA gives significant improvement on FID for tail classes, while maintaining head classes FID (even improve FID_head at initial training epochs.)
>
> **Q2: The mechanism by which $L_{al}$ implicitly extends or interacts with $L_{ucl}$ is not sufficiently explained... why UCL is not directly applied in the conditional latent space $e(*;c)$?**
>
> A2: Combining Alignment Loss and Unconditional Contrastive Loss on unconditional latents implicitly diversifies conditonal latents, as the reviewer has reaized. In the original submission, we have shown empirical improvement over directly applyng UCL to conditonal latent, which is coincidently the same as the concurrent work of Dispersive loss.
>
> We agree with the reviewer that a theoretical justification is needed, which is added to the revised version in Appendix A.5. It is known that InfoNCE loss for contrastive learning gives a lower bound of mutual information between data sample and representation/latents[1]. In other words, minimizing InfoNCE loss maximizes mutual information. We denote conditional latent as $h^c_{t} := e(x_t,t, c)$ and unconditional latent as $h^u_t := e_(x_t,t)$, where $c$ is the class condition. The InfoNCE loss on unconditional latents maximizes the mutual information $I(h_{t}^{u};x)$ between image $x$ and its unconditional latent $h_t^u$. Similarly, InfoNCE loss on conditional latents maximizes the mutual information $I(h_{t}^c;x,c)$ between image $x$ and its conditional latent $h_t^c$ and condition $c$.
>
> Based on the chain rule of mutual information, $I(h_{t}^c;x,c)$ can be decomposed into two pars:
>
> $I(h^{c}_t;x,c)=I(h^{c}_t;c)+I(h^{c}_t;x|c).$
>
> The first item $I(h^c_t;c)$ measures the mutual information between class $c$ and latent $h_t^c$, while the second item $I(x;h^c_{t}|c)$ measures the conditional mutual information. Intuitively, the first term tells how much class condition reveals about the latent, which reflects **inter-class diversity**. The second term measures conditional mutual information with condition $c$ which reflects **intra-class diversity**. To address the mode collapse issue for tail classes and increase diversity, it is apparent that we need to focus on maximizing the second item, i.e., conditional mutual information $I(h^{c}_{t}; x|c)$. However, it is likely that optimization is dominated by inter-class mutual information $I(h_t^c;c)$ which admits a trivial solution of having an identical latent for all images of the same class leading to maximum mutual information.
>
> We choose to diversify unconditional latents through InfoNCE loss with negative samples only, which maximizes $I(h^{}_t;x)$. In this case, there is no shortcut solution, and the model is forced to diversify all latents regardless of class. So, we choose to distill diversified unconditional latent to conditional latent with a simple alignment loss.
>
> Another practical benefit of our method is improving both conditional generation and unconditional generation, which are combined in classifier-free guidance during inference. Table 11 shows that CCUA also improved unconditional generation.
>
> **Q3: adapt Dispersive Loss to the long-tailed setting.**
>
> A3: We would like to point out that our $L_{ucl}$ can be seen as a variant of dispersive loss from the space of conditional latent to unconditional latent. Our ablation study in Table 4 for the hyper parameters shows that the joint loss is better than the $L_{ucl}$ alone.

---

> ### Author Response · Authors · 2025-11-23
> **Response 2/2 to Reviewer jyaR**
>
> **Q4: The rationale for treating all blocks’ outputs as the contrastive latent space, rather than using a bottleneck or intermediate representation**
>
> A4: We thank the reviewer for brining this up. We conduct ablation study to investigate the latent space selection in Table 12 of our revised submission. Please refer to our response to reviewer KAa1-Q2.
>
>
>
> **Q5: discuss or compare with more recent (2025 or latest) SOTA diffusion models**
>
> A5: Highly relevant to us is the concurrent work of dispersive loss for balanced diffusion model which we have compared to. We would be willing to compare to other recent work if the reviewer can point out specific recent paper.
>
> **Q6: whether including positive pairs (same-class samples) would negatively affect long-tailed generation, and whether relying solely on repulsive forces might lead to overly dispersed latent representations.**
>
> A6: We thank the reviewer for suggesting this variant with supervised contrastive learning[2], which we have considered. InfoNCE loss pulls positive pairs closer while pushing negative pairs far away. For long-tailed scene, we mainly focus on mode collapse issues for tail classes. If we include positive pair with samples from the same class, the mode collapse issue would be more severe for tail classes since the infoNCE loss would compress the tail classes representations in the latent space.
>
> We admit that relying solely on repulsive forces leads to overly dispersed latent representations and large recall as shown in our ablation study of loss weights in Table 4. Thus we apply the timestep weight to the proposed alignment loss. For a small timestep, the representations are more constrained by the class condition and thus we align less on the conditional-unconditional space.
>
>
> [1] Oord, Aaron van den, Yazhe Li, and Oriol Vinyals. "Representation learning with contrastive predictive coding." arXiv preprint arXiv:1807.03748 (2018).
>
> [2] Khosla, Prannay, et al. "Supervised contrastive learning." Advances in neural information processing systems 33 (2020): 18661-18673.

---

### Official Review · Reviewer_2xLz · 2025-10-27

**Soundness:** 2
**Presentation:** 3
**Contribution:** 2
**Rating:** 6
**Confidence:** 3

**Summary:**

This paper studies the problem of long-tailed image generation with diffusion models. The authors observe that diffusion models trained on long-tailed datasets (e.g., ImageNet-LT) tend to produce low-quality and less diverse images for tail classes. To address this, they propose a new regularization framework called CCUA (Contrastive Conditional–Unconditional Alignment), which combines (1) an Unsupervised Contrastive Loss (UCL) to promote feature dispersion and prevent latent collapse, and (2) an Alignment Loss (AL) to align conditional and unconditional noise predictions in the early timesteps of diffusion, enabling structural knowledge transfer from head to tail classes. Extensive experiments demonstrate that the proposed framework improves the generation quality of diffusion models.

**Strengths:**

- The paper effectively integrates multiple techniques to address the long-tailed generation problem. The approach is well motivated, drawing inspiration from prior research on class-imbalance GANs, and demonstrates empirical effectiveness.
- The method consistently performs well across both UNet-based architectures and Diffusion Transformers (DiT), highlighting its generalization capability.
- The paper is well-structured, with intuitive visualizations illustrating improved tail-class diversity and fidelity.

**Weaknesses:**

- As acknowledged in the paper, the training process requires a longer duration.
- Loss-function ablation is evaluated only with FID, lacking complementary metrics (IS, Precision/Recall, FID-tail) to substantiate the claimed diversity–fidelity balance.
- While the results are promising, it remains somewhat unclear how much of the improvement stems from the proposed CCUA loss compared to the batch resample strategy. Since resampling already provides gains with minimal cost, a controlled comparison isolating CCUA’s effect would further clarify its independent contribution and better justify the added training overhead.

**Questions:**

- In the ablation study on the loss function, it would be helpful to include additional evaluation metrics such as Inception Score (IS) and Precision/Recall to provide a more comprehensive analysis.
- Would it be possible to include an experiment similar to Table 5 on the TinyImageNet-LT dataset to further validate the method’s generalization?

---

> ### Author Response · Authors · 2025-11-23
>
> We thank reviewer 2xLz who finds our paper to be well-structured, well-motivated and showing generalization capabilities. We address the concerns bellow.
>
> **Q1: As acknowledged in the paper, the training process requires a longer duration.**
>
> A1: Yes, our method incurs training overhead thought it doesn't increase inference latency. We have further optimized our implementation which technicaly gives the same gradients and results. The training overhad is reduced to **1.3x** on SiT with our accelerated impelemntation. Please refer to our response to Q1 of Reviewer KAa1 and Appendix A.2 for our acceleration trick.
>
> **Q2: Loss-function ablation is evaluated only with FID, lacking complementary metrics (IS, Precision/Recall, FID-tail) to substantiate the claimed diversity–fidelity balance.**
>
> A2: We thank the reviewer for suggesting additional metrics for the ablation study. Please find additinal metrics in our response to Q3 for reviewer KAa1 and Table 4 of hyper parameters analysis in our revised submission. These metrics including IS, Precision/Recall metric indeed help us trade off diversity and fidelity and select weights for loss functions. It also shows the two losses are complementary.
>
> **Q3: Would it be possible to include an experiment similar to Table 5 on the TinyImageNet-LT dataset to further validate the method’s generalization?**
>
> A3: We conduct Batch Resample ablation on TinyImageNet-LT dataset as shown in Table 6 in our revised submission, which shows consistent improvement with the ablation on ImageNet-LT in Table 5. The proposed CCUA losses achieves noticeably better performance compared to the baseline no matter with or without batch resample strategy.
>
> | Method | FID | FID$_{tail}$ | KID |
> | - | - | - | - |
> | DDPM (baseline) | 18.67 | 40.12 | 6.31 |
> | +Batch Resample | 17.18 (-1.49) | 33.60 (-6.52) | 4.66 (-1.65) |
> | $\mathcal{L}_{ccua}$ | 17.16 (-1.51) | 37.88 (-2.24) | 4.89 (-1.42) |
> | +Batch Resample | 15.24 (-3.43) | 30.39 (-9.73) | 3.83 (-2.48) |

---

> > ### Comment · Reviewer_2xLz · 2025-11-28
> >
> > Thanks for the author's response. My concerns have been resolved, and I will keep my score.

---

### Official Review · Reviewer_yVZe · 2025-10-29

**Soundness:** 2
**Presentation:** 2
**Contribution:** 2
**Rating:** 2
**Confidence:** 4

**Summary:**

The paper introduces CCUA, a framework designed to improve the performance of class-conditional diffusion models trained on long-tailed datasets. CCUA combines unsupervised contrastive loss and alignment loss together to address the problem that models trained on imbalanced data tend to suffer from mode collapse and reduced fidelity for tail classes. The method shows empirical enhancement on ImageNet-LT,Places-LT and CIFAR-LT variants.

**Strengths:**

1. The two proposed losses are intuitive and well-motivated and the method is also simple to implement, plus easy to integrate into existing diffusion pipelines.

2. The empirical evidence is extensive: results span multiple strong baselines, architectures (U-Net, DiT), and datasets. CCUA consistently outperforms prior work, especially in terms of FID for tail classes and diversity (Recall), as seen in Table 1, Table 2, and Table 3.

**Weaknesses:**

The primary weakness is that the paper's contribution appears to be an assemblage of existing or concurrent ideas rather than a fundamental innovation. The first component, UCL, is a dispersive loss. The authors themselves state it is "similar to" concurrent work on dispersive losses. The second component, AL, is admittedly a direct adaptation from the long-tailed GAN literature. The paper's insight is to translate the "low-resolution alignment" from GANs to "large timestep alignment" in diffusion, which is a reasonable but straightforward adaptation. Therefore, the core contribution is limited to combining these two specific losses into an integrated loss, which feels more incremental than foundational.

The paper provides insufficient justification and no deep mathematical analysis for why this specific combination of UCL and AL is optimal. The two losses are simply added together with weighting hyperparameters. There is no deeper analysis or theoretical argument for how these two losses interact or why they form a principled, unified framework, beyond the high-level intuition.

The method introduces a major practical drawback. The paper's "Limitation" section (Appendix, page 15) explicitly states the method increases training time significantly (1.6x for DDPM, 1.48x for SiT) due to the two forward passes required for the Alignment Loss.
Furthermore, UCL (a contrastive-style loss) typically benefits from large batch sizes to see a rich set of negatives. However, the 1.5-1.6x increase in computational cost from the AL component would likely force a reduction in batch size. The paper fails to analyze or discuss this practical tension.

The paper's empirical claims are not sufficiently stress-tested, which is a significant flaw for a method that relies on a weighted-loss combination. The paper completely lacks a sensitivity analysis for its two most critical hyperparameters, $\alpha$ and $\gamma$. It is impossible to know if the strong results are due to extensive, unreported tuning or if the method is robust. The robustness analysis is very limited. It only tests robustness to random seeds (Appendix Table 10) and fails to analyze performance across different imbalance factors to see how it performs as the long-tail problem becomes more or less severe.The evaluation metrics, while good, are limited. For generative models, metrics like CLIP Score are often vital for judging image-concept alignment, especially for diverse tail classes, but this is missing from the evaluation.

The paper fails to adequately discuss or compare against highly relevant prior work on contrastive diffusion (e.g., Yan et al., 2022) and other conditional diffusion models (Wang et al., 2023).

**Questions:**

See weekness

---

> ### Author Response · Authors · 2025-11-23
> **Response to Reviewer yVZe**
>
> We appreciate the critical feedback from Reviewer yVZe.
>
> **Q1: The primary weakness is that the paper's contribution appears to be an assemblage of existing or concurrent ideas rather than a fundamental innovation. The first component, UCL, is a dispersive loss... The second component, AL, is admittedly a direct adaptation from the long-tailed GAN literature.**
>
> A1: We respectively disagree with the reviewer that we are just combining existing losses. Our UCL loss on unconditional generation is similar but not the same as dispersive loss on conditional generation. Additionally, according to the reviewer/AC guideline, authors are encouraged to discuss but not required to compare to contemporaneous work or unpublished arxiv papers. We discussed dispersive loss paper in detail and even provided experimental comparison in Table 1 in the original submission. Our alignment loss is also novel as it hasn't been explored for diffusion model. The combination of the two losses is principled, see next response.
>
> **Q2: The paper provides insufficient justification and no deep mathematical analysis for why this specific combination of UCL and AL is optimal.**
>
> A2: Here we clarify that the two losses are not independent. UCL diversifies unconditonal latent and AL aligns conditional latent with unconditinal latent, which implicitly diversify conditional latent.
>
>
> We choose not to directly diversify conditional latents, as is done in concurrent work of Dispersive Loss. Our combined loss is more effective for addressing overfitting in tail classes empirically verified in Table 1. The geometric interpretation for the reason is that the diversity of conditional latents involves both inter-class variance and intra-class variance. Directly applying InfoNCE loss on conditional latent can lead to a shortcut solution that dominantly maximizes inter-class variance. However, contrastive regularization on unconditional latents *must* diversity latents for all data regardless of class. Aligning conditional latents to diversified unconditional latents via distillation is more effective. In Appendix A.5 of our revised submission, we give a theoretical analysis, which shows maximizing mutual information between data and uncoditional latents through InfoNCE is a better way to diversify latents and increase diversity for tail classes.
>
> Another benefit of such combination is improvement for both conditional and unconditional generation, which are used in classifier-free guidance during inference.
>
> **Q3: the method increases training time significantly (1.6x for DDPM, 1.48x for SiT) due to the two forward passes required for the Alignment Loss... the 1.5-1.6x increase in computational cost from the AL component would likely force a reduction in batch size.**
>
> A3: We agree this is a practical drawback. We further optimized our  implementation with a simple trick to reduce training time. Please find our response to Q1 for Reviewer KAa1. With the accelerated implementaion, our method is 1.3x slower than SiT with the same batch size.
>
> **Q4: The paper completely lacks a sensitivity analysis for its two most critical hyperparameters, $\alpha$ and $\gamma$... fails to analyze performance across different imbalance factors...The evaluation metrics, while good, are limited. For generative models, metrics like CLIP Score are often vital...**
>
> A4: We provide hyper-parameters experiments for sensitivity analysis of the proposed two losses. Please refer to our reponse to Reviewer KAa1-Q3.
>
> For CLIP Score, this paper is focused on class-conditioned image generation instead of text-to-image. It is not a common practice to measure CLIP score for class-conditioned image generation.
>
>
> **Q5: The paper fails to adequately discuss or compare against highly relevant prior work on contrastive diffusion (e.g., Yan et al., 2022) and other conditional diffusion models (Wang et al., 2023).**
>
> A5: We appreciate if the reviewer can provide more details, i.e., titles of these paper mentioned. We are not sure which papers the reviewers are referring to (Yan et al., 2022 and Wang et al., 2023).

---

### Official Review · Reviewer_KAa1 · 2025-10-31

**Soundness:** 2
**Presentation:** 2
**Contribution:** 2
**Rating:** 4
**Confidence:** 4

**Summary:**

This paper addresses the challenge of training class-conditional diffusion models on long-tailed datasets, where tail classes suffer from poor sample quality and diversity (mode collapse) due to data scarcity. The authors propose a framework named Contrastive Conditional–Unconditional Alignment (CCUA), which introduces two novel loss functions to the standard diffusion model training pipeline. The authors demonstrate the effectiveness of CCUA by integrating it into both U-Net and Diffusion Transformer (DiT/SiT) architectures. Extensive experiments on long-tailed versions of CIFAR, TinyImageNet, Places, and ImageNet show that CCUA significantly improves generation quality (FID) and diversity (Recall) for tail classes, often without compromising (and sometimes improving) the quality of head classes.

**Strengths:**

- The paper tackles the practical problem of generative model performance on imbalanced, real-world data. The proposed method is simple and intuitive.
- The method is validated on a wide range of standard long-tailed datasets (ImageNet-LT, Places-LT, TinyImageNet-LT, etc.) at various resolutions, including challenging 256x256 generation.
- The paper is well-written, clearly structured, and easy to follow.

**Weaknesses:**

1. Training Cost: The alignment loss (Lal) requires a forward pass for both the conditional and unconditional predictions for each sample during training. The appendix (line 802) states this results in a ~1.6x increase in training time per epoch compared to a standard DDPM. This is a significant computational overhead that should be mentioned and discussed as a limitation in the main paper, not just in the appendix.

2. Justification for Latent Space h: The definition of the encoder e(*) that produces the latent h for the contrastive loss feels somewhat arbitrary, especially for the Diffusion Transformer. Some justification for why this specific feature level is optimal for applying the repulsive force would strengthen the paper. It is unclear if other layers were considered.

3. Unclear Rationale for Hyperparameter Choices:
- The loss weighting parameters $\alpha$ and $\lambda$ are introduced but their values, selection process, and sensitivity are not discussed.
- The appendix (line 723) mentions that for TinyImageNet-LT and Places-LT, the UCL was also weighted by t/T, similar to the AL. This seems like a critical implementation detail. The rationale for applying this timestep weighting to UCL for some datasets but not others is not provided, which raises questions about the method's generality and the tuning required for new datasets.

**Questions:**

1. Regarding the training overhead, you mention a 1.6x increase in training time in the appendix. Could you please discuss this limitation in the main paper? Is it possible that the improved convergence speed (e.g., reaching a better FID in fewer epochs, as seen in Table 1) partially amortizes this per-epoch cost? What's the performance gap between baselines and CCUA given the same training budget measured in GPU$\times$Day?

2. Could you elaborate on the design choice for the latent representation h used in the UCL? Specifically for the Diffusion Transformer, why was the output of the final transformer block chosen? Did you experiment with applying the contrastive loss at different depths of the network, and how did that affect performance? Moreover, could you explain why

3. In Table 3 (TinyImageNet-LT), CCUA shows a slightly higher FID on 'Head' classes compared to some baselines. Could you comment on this small performance degradation and whether there is a fundamental trade-off between improving tail classes and maintaining head class performance?

4. The alignment loss L_al is linearly weighted by t/T. Have you experimented with other weighting schedules (e.g., quadratic, or a step function that applies the loss only for t > T_threshold)? How sensitive is the model's performance to this specific linear schedule?
In the appendix (line 723), you note that the UCL was also weighted by t/T for the TinyImageNet-LT and Places-LT datasets. This detail is not mentioned in the main method description. Could you clarify why this was necessary for these specific datasets and not for ImageNet-LT? This seems to suggest the method's components might require dataset-specific tuning.

5. How's the performance of CCUA w.r.t. the imbalanced factor (e.g., 0.001 as in OCLT)?

---

> ### Author Response · Authors · 2025-11-23
> **Response 1/2 to Reviewer KAa1**
>
> We thank reviewer KAa1 for the through review. We provide responses to questions and raised weaknesses as follows.
>
> **Q1: This is a significant computational overhead that should be mentioned and discussed as a limitation in the main paper... What's the performance gap between baselines and CCUA given the same training budget measured in GPU Day?**
>
>
> A1: We thank the reviewer for pointing out this limitation. We further optimized our implementation by a simple trick of computing conditional generation and unconditional generation for the same batch with one function call of model.forward(). In our original implementation, we called model.forward() twice in each iteration, which is not as efficient.
>
> ```python3
> # x_batch.shape: B, C, H, W
> # original implementation for one iteration
> cond_output = model.forward(x_batch, c)
> uncond_output = model.forward(x_batch, null)
>
> # optimized implementation for one iteration
> cond_output, uncond_output = model.forward(torch.cat([x_batch, x_batch], dim=0), torch.cat([c, null], dim=0)).chunk(2, dim=0)
> ```
>
> At the cost of increased GPU memory consumption, our optimized implementation is only **1.3x** slower than SiT with original diffusion loss. We have included Section 4.4 Limitation of our method in the main paper.
>
> | Method | Average Training Time (steps per second) | GPU Consumption (GB per image) |
> | ------ | --------------------- | --------------- |
> | SiT      | 8.7 | 0.44 |
> | CCUA (original implementation) | 5.8 | 0.48 |
> | CCUA (accelerated implementation)   | 6.7 | 0.52 |
>
>
> We measure our method on the same training time with SiT, which is reported in the last row of Table 1 in our revised submission. With the same training days (160 epochs for vallina SiT), CCUA still achieves the better IS, sFID score while increasing the fidelity as relected in lower FID (from 19.9 to 17.2), and better diversity as reflected in higher Recall (from 18.6 to 21.6).
>
> | Method | IS | FID | sFID | Prec. | Recall |
> | - | - | - | - | - | - |
> | SiT | 111.7 | 19.9 | 20.1 | 70.3 | 18.6 |
> | CCUA | 119.0 | 17.2(-2.7) | 13.9 | 68.4 | 21.6(+3.0) |
>
> **Q2: The definition of the encoder e(\*) that produces the latent h for the contrastive loss feels somewhat arbitrary, especially for the Diffusion Transformer.**
>
> A2: We conduct the ablation study to investigate the choice of latent encode network in SiT. For a SiT model with $N$ SiT/DiT blocks, we use the latent representations $h$ from the $N/4$-th, $N/2$-th, $3N/4$-th, and the $N$-th block for $\mathcal{L}_{ucl}$ loss. As shown in Table 12 in our revised submission, we found even the last layer does not always provide the best performance on all metrics, it has the best spatial FID (sFID) score and Recall. These two metrics reflect the better quality and diversity of representations in latent space, thus we select the N-th layer, i.e. the last SiT block, as the encoder layer.
>
> | Epochs | Latent | IS$\uparrow$ | FID$\downarrow$ | sFID$\downarrow$ | Prec.$\uparrow$ | Recall$\uparrow$ |
> | - | - | - | - | - | - | - |
> | 40 | N/4 | 70.69 | 27.04 | 20.50 | **60.16** | 20.12 |
> |    | N/2 | 68.57 | 28.00 | 21.13 | 58.59 | 21.28 |
> |    | 3N/4 | 68.48 | 27.99 | 19.96 | 58.96 | 21.09 |
> |    | N | **73.60** | **25.89** | **16.54** | 58.62 | **23.05**  |
> | 80 | N/4 | **111.91** | **19.43** | 18.25 | **69.44** | 18.88 |
> |  | N/2 | 105.41 | 20.10 | 19.49 | 68.22 | 19.24 |
> |  | 3N/4 | 109.29 | 19.78 | 18.50 | 69.30 | 18.80 |
> |  | N | 103.66 | 19.99 | **14.88** | 65.80 | **21.31** |
> | 120 | N/4 | **140.46** | **16.29** | 17.19 | **73.92** | 18.40 |
> |  | N/2 | 134.89 | 16.63 | 17.89 | 72.94 | 18.86 |
> |  | 3N/4 | 137.21 | 16.65 | 17.46 | 73.60 | 18.41 |
> |  | N | 119.11 | 17.55 | **13.89** | 68.38 | **21.20** |
> | 160 | N/4 | **153.07** | **15.08** | 16.54 | **75.73** | 17.27 |
> |  | N/2 | 148.61 | 15.14 | 16.94 | 75.40 | 18.11 |
> |  | 3N/4 | 148.88 | 15.22 | 16.80 | 75.32 | 19.35 |
> |  | N | 124.87 | 16.41 | **13.17** | 69.84 | **21.34** |

---

> ### Author Response · Authors · 2025-11-23
> **Response 2/2 to Reviewer KAa1**
>
> **Q3: Unclear Rationale for Hyperparameter Choices.**
>
> A3: We agree such abltion is necessary. Table 4 of our revised submission gives ablation of loss weights. Using extremely large $\alpha=1$ and $\gamma=1$ causes much higher Recall and lower Precision, which means better diversity but worse fidelity. We selected the optimal ($\alpha=0.05$, $\gamma=0.05$) for the best FID while trading off diversity and fidelity.
>
> | Method | $\alpha$ | $\gamma$ | IS$\uparrow$ | FID$\downarrow$ | sFID$\downarrow$ | Prec.$\uparrow$ | Recall$\uparrow$ |
> | ------ | -------- | -------- | -- | --- | ---- | ----- | ------ |
> | SiT | - | - | 53.93 | 33.87 | 22.66 | 54.56 | 19.17 |
> | CCUA (ours) | 1.0 | 1.0 | 52.54 | 32.96 | **16.26** | 48.92 | **29.90** |
> | CCUA (ours) | 0.5 | 0.5 | 62.26 | 30.07 | 20.32 | 55.55 | 22.29 |
> | CCUA (ours) | 0.05 | 0.05 | **73.60** | **25.89** | 16.54 | 58.62 | 23.05 |
> | CCUA (ours) | 0.01 | 0.01 | 65.79 | 28.61 | 18.97 | 58.52 | 19.75 |
> | CCUA (ours) | 0.05 | 0 | 64.28 | 28.70 | 18.69 | 57.54 | 21.25 |
> | CCUA (ours) | 0 | 0.05 | 64.65 | 29.04 | 20.17 | **58.64** | 19.97 |
>
> **Q4:  How sensitive is the model's performance to this specific linear schedule?... The rationale for applying this timestep weighting to UCL for some datasets but not others is not provided.**
>
> A4: We clarify that we applied the timestep weighting by t/T to UCL for all datasets, not only for TinyImageNet-LT and Places-LT. We have revised our manuscript. We thank the reviewer for suggesting other weight scheduler, for which we will report results in future versions.
>
> **Q5: CCUA shows a slightly higher FID on 'Head' classes compared to some baselines. Could you comment on this small performance degradation and whether there is a fundamental trade-off between improving tail classes and maintaining head class performance?**
>
> A5: We agree with the reviewer that it can be a concern for tail class. We further tested FID-head and FID-tail on ImageNet-LT for SiT, and include these metrics in Table 1 of our revised submission.
>
> | Epochs | Method | FID$_{head}\downarrow$ | FID$_{tail}\downarrow$ |
> | - | - | - | - |
> | 40 | SiT | 23.4 | 52.8 |
> |    | CCUA | 21.6 (-1.8) | 28.1 (-24.7) |
> | 80 | SiT | 18.5 | 41.6 |
> |    | CCUA | 17.2 (-1.3) | 23.0 (-18.6) |
> | 120 | SiT | 17.5 | 35.5 |
> |    | CCUA | 17.5 | 20.0 (-15.5) |
> | 160 | SiT | 17.9 | 33.9 |
> |    | CCUA | 17.9 | 19.6 (-14.3) |
>
> Compared to SiT, CCUA gives significant improvement on FID (e.g., from 52.8 to 28.1 for 40 epcohs) for tail classes, while maintaining head classes FID (even improve the head classes FID from 23.4 to 21.6 at initial training epochs.)
>
> **Q6: performance of CCUA w.r.t. the imbalanced factor?**
>
> A6: We thank the reviewer for suggesting this experiment. We tested CCUA with SiT using imbalanced factor 0.001 for ImageNet-LT, shown in Table 8 in our revised submission. For ImageNet datasets, each class only contains 1300 images. With 0.001 imbalanced factor the tail classes only contains 1~2 images. Our method improves the baseline SiT significantly for such a challenging dataset.
>
> | Epochs | Method | IS | FID | sFID | Prec. | Recall |
> | - | - | - | - | - | - | - |
> | 40 | SiT | 29.89| 51.26 | 21.26 | 38.09 | 22.04 |
> |    | CCUA | 48.79 | 36.41 (-14.85) | 16.41 | 45.04 | 21.26 |
> | 80 | SiT | 46.47 | 37.55 | 23.50 | 48.66 | 20.39 |
> |    | CCUA | 83.93 | 25.96 (-11.59) | 17.44 | 55.91 | 19.33 |
> | 120 | SiT | 59.05 | 30.56 | 22.05 | 54.79 | 20.61 |
> |    | CCUA | 105.86 | 22.44 (-8.12) | 18.40 | 60.34 | 19.29 |
> | 160 | SiT | 67.70 | 27.25 | 19.96 | 57.49 | 18.89 |
> |    | CCUA | 118.16 | 20.93 (-6.32) | 18.88 | 62.88 | 18.89 |

---

> ### Comment · Reviewer_KAa1 · 2025-11-27
>
> Thank you for your detailed replies. I have a few remaining concerns regarding the experiment:
>
> **New Q1:** In your response to Q1, when compared to SiT with equivalent training days, the FID improvement from CCUA may stem from the additional supervising signal provided by the contrastive loss in Eq 1 accelerating the overall training process. This raises a critical question: is the primary benefit of CCUA a fundamental improvement in handling long-tailed data, or is it a training acceleration technique that helps DM learn more general representations and could potentially be matched by extending the baseline's training time or using the latest training techniques (e.g., REPA published in 2024)?
>
> **New Q2:** Meanwhile, could you provides the FID on head and tail classes when trained for the same time as the SiT baseline in Table 1?
>
> **New Q3:** **In the main paper, it's important to provide a more straightforward, fair, and apple-to-apple comparison with baseline methods thus clearly positions the method's contribution within the literature.** I've noted an inconsistency in the results that raises questions about fairness:
> - On ImageNet-LT, your method performs on par with that reported in CBDM[1] (FID ≈ 16.3, Table 7).
> - However, on TinyImageNet-LT 64x64 and Places-LT 64x64 (Table 2), your method shows a much larger performance advantage.
>
> Why is the performance gap so much larger on these smaller datasets? This discrepancy suggests the comparisons in Table 2 may not be fair. Please justify this difference or ensure the experimental setup for Table 2 is a true apples-to-apples comparison.
>
> [1] Qin, Yiming, et al. "Class-balancing diffusion models." Proceedings of the IEEE/CVF Conference on Computer Vision and Pattern Recognition. 2023.

---

> ### Author Response · Authors · 2025-12-03
> **Response to New Questions**
>
> We thank reviewer KAa1 for raising additional questions.
>
> **Response to new Q1:** We argue that our method of CCUA yields fundamental improvement in better representation for image generation in particular for tail classes. The benefit is not limited to training acceleration. To see this, we further trained models for 240 epoches on ImageNetLT dataset. The table bellow shows CCUA still outperforms SiT baseline by a large margin in terms of overall FID (17.8 v.s. 14.6). Note that while FID_head has converged for 240 epochs, our method gives dramatic improvment of FID_tail over SiT baseline. With more training epochs, SiT baseline tends to overfit more on classes with limited data. Our CCUA regularization losses mitigates such issues of overfitting.
>
> For the same number of epochs, CCUA also outperms REPA, which is the latest training technique suggested by the reviewer.
>
> **Table 1. Comparison on ImageNet-LT 256×256 with SiT pipeline.**
> | Epoch | Method | IS$\uparrow$ | FID$\downarrow$ | sFID$\downarrow$ | Prec.(\%)$\uparrow$ | Recall(\%)$\uparrow$ | FID$_{head}\downarrow$ | FID$_{tail}\downarrow$ |
> | - | - | - | - | - | - | - | - | - |
> | 40 | SiT  | 53.9 | 33.8 | 22.6 | 54.5 | 19.1 | 23.3 | 52.8 |
> |    | CBDM | 54.8 | 34.1 | 23.3 | 53.9 | 18.7 | 23.3 | 53.1 |
> |    | REPA | **74.1** | 28.4 | 20.0 | 58.3 | 16.1 | **17.8** | 48.5 |
> |    | **CCUA (ours)** | 73.6 | **25.8** | **16.5** | **58.6** | **23.1** | 21.5 | **28.1** |
> | 120 | SiT  | 103.1 | 21.2 | 20.1 | 69.3 | 18.2 | 17.5 | 35.4 |
> |     | CBDM | 105.7 | 20.9 | 20.7 | **70.3** | 17.8 | 17.7 | 35.5 |
> |     | REPA | **126.8** | 19.7 | 20.3 | 68.0 | 15.8 | 17.7 |36.9 |
> |     | **CCUA (ours)** | 119.1 | **17.5** | **13.8** | 68.3 | **21.2** | **17.4** | **20.0**
> | 160 | SiT  | 111.7 | 19.9 | 20.1 | 70.3 | 18.6 | 17.8 | 33.9 |
> |     | CBDM | 117.2 | 19.4 | 20.2 | **72.5** | 17.7 | 17.9 | 32.7 |
> |     | REPA | **137.8** | 18.1 | 19.3 | 69.8 | 16.2 | 18.0 | 33.8 |
> |     | **CCUA (ours)** | 124.8 | **16.4** | **13.1** | 69.8 | **21.3** | **17.8** | **19.5** |
> | 240 | SiT | 132.2 | 17.8 | 19.6 | **74.4** | 17.6 | 18.8 | 29.6 |
> |     | **CCUA (ours)** | **141.8** | **14.6** | **13.2** | 73.5 | **19.6** | **18.5** | **21.6** |
>
> **Response to New Q2:** We have added FID_head and FID_tail for CCUA with the same training time as SiT baseline. SiT achieves 17.8 FID_head and 33.9 FID_tail, while CCUA achieves 17.5 FID_head and 20.1 FID_tail with the same training time. This demonstrates CCUA's capability of improving tail class images while maintaining head class.
>
> **Response to New Q3:** The pioneering work of CBDM[1] has a different experiment settings with our ImageNetLT experiments in Table 1.
>
> Firstly, as mentioned in [Table 7 of CBDM[1]](https://arxiv.org/pdf/2305.00562), 'Here, CB-DDPM refers to the DDPM model fine-tuned by our method. We note that the fine-tuning is applied on ImageNet-LT due to the limited calculation budget.', the CBDM's reported FID=16.3 is actually a DDPM pretrained model **further fintuned by CBDM**.
> In our experimental settings of Table 1 (ImageNetLT) and Table 2 (TinyImageNetLT and PlacesLT), we train all models from scratch to the same number of epochs to keep the fairness of the experiments.
>
> Secondly, CBDM[1] reported numbers on ImageNetLT is based on DDPM backbone while our experiment on ImageNetLT is based on SiT backbone, which are not comparable. Though the CBDM[1] authors didn't provide such implementation that applying CBDM to SiT framework, we report CBDM with SiT backbone in our revised Table 1 based our re-implementation. As shown in revised Table 1, CCUA achieves better results than CBDM at the same experimental settings.
>
>
> [1] [Qin, Yiming, et al. "Class-balancing diffusion models." Proceedings of the IEEE/CVF Conference on Computer Vision and Pattern Recognition. 2023.](https://arxiv.org/pdf/2305.00562)

---

### Author Response · Authors · 2025-12-04
**Rebuttal and Revision Summary**

Dear reviewers and area chairs,

We sincerely thank all the Reviewers for their constructive and insightful feedback..


**Strengths**

- We appreciate reviewers KAa1, yVZe, and 2xLz for recognizing that the proposed UCL and AL losses are **simple, intuitive, well-motivated**, and easy to integrate into existing diffusion pipelines.
- We thank all reviewers for acknowledging our **consistent empirical improvements** across datasets and architectures, as well as noting that CCUA effectively addresses the practical and **important real-world challenge of mode collapse** in **long-tailed class-conditional diffusion**.
- We also appreciate the positive comments on the clarity and organization of the paper.


**Weaknesses**

The main concerns from reviews centered on

- **Theoretical Justification** for why UCL and AL form a principled combination rather than two independent losses (yVZe, jyaR).
- **Training Efficiency**, as CCUA introduces additional training overhead though inference is unchanged (KAa1, yVZe, 2xLz).
- **Fair Comparison** against SiT and recent state-of-the-art long-tailed generative methods (KAa1).
- **Latent Space Choice and Hyperparameter Sensitivity**, viewed as under-justified in the initial submission (KAa1, yVZe, 2xLz).

We believe that our added clarifications, expanded experiments, and point-by-point responses adequately resolve all raised concerns.

**Summary of Revision**

- **Theoretical Explanation.** We added a detailed mutual-information analysis (Appendix A.5) explaining why applying InfoNCE directly on conditional latents can lead to shortcut solutions dominated by inter-class separation, thereby amplifying tail-class collapse. In contrast, UCL on unconditional latents maximizes $I(h_t^u;x)$ without shortcuts, and AL naturally transfers this increased intra-class diversity to the conditional latents. This formally justifies why UCL + AL is a principled and more effective combination rather than an ad-hoc sum of losses (addressing concerns from yVZe and jyaR).
- **Efficient Implementation of CCUA.** We introduced an accelerated implementation that merges the conditional and unconditional forward passes into a single batched call. This reduces training cost from 1.6× to 1.3× relative to SiT, with identical gradients and unchanged inference speed. This optimization is now included in the revised paper and directly addresses efficiency concerns raised by KAa1, yVZe, and 2xLz.
- **Additional Experiments**
    - **Same Training Time CCUA with SiT.** We showed that under identical training budgets (same epochs and GPU days), CCUA both significantly improves FID and Recall over SiT. (addressing KAa1).
    - **Head/Tail Metrics Added for ImageNet-LT.** We now report both FID_head and FID_tail. CCUA substantially improves FID_tail while matching or slightly improving FID_head, supporting our claim that CCUA boosts tail diversity without sacrificing head-class fidelity (addressing KAa1, jyaR).
    - **Experiments Under Extreme Imbalance (factor = 0.001).** With only 1–2 images per tail class, CCUA still achieves large improvements over SiT across IS, FID, and Recall. This demonstrates robustness under even more challenging long-tail regimes (addressing KAa1, yVZe).
    - **Extra Comparison to SOTA methods on ImageNetLT dataset.** We conducted further comparisons against recent state-of-the-art long-tailed diffusion methods, including CBDM, and the latest training technique REPA. Our extended ImageNet-LT results show that CCUA consistently outperforms SiT, CBDM, and REPA across all training schedules, and the gains persist even when training is extended to 240 epochs, confirming that CCUA’s improvements arise from fundamentally better representation learning rather than training acceleration. Finally, we clarified that CBDM’s previously reported FID=16.3 is based on DDPM fine-tuning, not training-from-scratch, and thus not directly comparable; our re-implementation of CBDM on SiT ensures a fully fair evaluation, where CCUA remains clearly superior.
- **Ablation Study**
    - **Latent Space Choice.** We added a full latent space selection ablation across SiT layers (N/4, N/2, 3N/4, N). The last block consistently yields the best sFID and Recall, providing empirical grounding for our encoder selection (addressing KAa1, jyaR).
    - **Hyperparameter Experiments.** We conducted comprehensive studies on $\alpha$ and $\gamma$ across IS/FID/sFID/Precision/Recall metrics. Results reveal a clear diversity–fidelity trade-off and justify the choice $\alpha=\gamma=0.05$ as an optimal setting (addressing KAa1, yVZe, 2xLz).


**Lastly**, we have empirically shown the advantage of our method over the concurrent work of Dispersive Loss in our original submission. We have discussed in detail the difference between CCUA and Dispersive Loss and given theoretical insights on why CCUA is more suitable than dispersive loss for long-tailed generation.

---

### Meta-Review · Area_Chair_Z3ro · 2026-01-05

**Summary:**

This paper initially received mixed reviews: one borderline accept (6, 2xLz), two borderline rejects (4, KAa1 & jyaR), and one reject (2, yVZe). The reviewers recognized the practical value of the proposed problem setting, the simple yet effective and versatile method, performance improvement, extensive experiments, and clarity of the paper. They however also raised multiple critical concerns with high training cost (KAa1, yVZe, 2xLz), incremental novelty (yVZe), lack of theoretical grounding (yVZe, jyaR), insufficient experimental verification (KAa1, 2xLz), lack of justification for the latent space selection (KAa1, yVZe), unclear rationale for the hyperparameter setting (KAa1, yVZe), missing discussion regarding the trade-off between tail and head classes (KAa1), insufficient performance analysis (KAa1, yVZe), missing comparisons with latest work (KAa1, yVZe), and evaluation with a limited set of performance metrics (2xLz, jyaR). The authors responded to these comments through the rebuttal and revision, but failed to assuage all of them, in particular, those about the expensive training cost, lack of theoretical grounding, lack of justification for the latent space selection, unclear rationale for the hyperparameter setting, the trade-off between tail and head classes, and insufficient performance analysis. The AC found the concerns that have not been successfully addressed outweigh the positive comments and the rebuttal, and thus regrets to recommend rejection. The authors are encouraged to reflect the reviewers' valuable comments and submit to next venues.

**Reviewer Concerns:**

[Concerns that have not been fully resolved]
- High training cost (KAa1, yVZe, 2xLz)
	- *During the discussion period, the authors reduced the training time at the cost of additional GPU memory footprint by re-implementing their method, the improved version is still 1.3 times slower than the original diffusion counterpart and demands more GPU memory though. A positive point is that the proposed method outperformed the diffusion baseline in terms of generation quality when trained using the same computation budget. The AC considers this concern has not been fully resolved since the proposed method still demands more computation and GPU memory even with the improved implementation.*
- Lack of theoretical grounding (yVZe, jyaR)
	- *The reviewer yVZe pointed out the lack of theoretical and empirical justification for why the proposed method is optimal for the purpose. The rebuttal and revision seem not well address this issue as no solid mathematical background or logic was given for the newly added analysis. For example,*
		- *The argument "it is likely that the optimization is dominated by inter-class mutual information" in line 1214 has not been guaranteed.*
		- *The second part of Sec. A.5 needs to be elaborated on; it is currently too short to clearly verify the proposed method.*
	- *The reviewer jyaR commented that the interaction between the two loss functions, which was claimed in the paper, has not been verified clearly. The authors addressed this issue together with that of the reviewer yVZe in Sec. A.5, but as aforementioned, the theoretical analysis looks not thorough enough.*
- Insufficient performance analysis (KAa1, yVZe)
	- *The reviewer KAa1 and yVZe asked further analysis on the performance of the proposed method versus the imbalance factor. The authors presented additional experimental results in the rebuttal and demonstrated that their method achieves strong performance even with an extreme imbalance factor (0.001). However, unfortunately, the comparison was done with only one baseline, unlike the other experiments.*
	- *The reviewer KAa1 was also wondering the impact of the loss weight schedule. the authors did not address this issue at all.*
- Lack of justification for the latent space selection (KAa1, yVZe)
	- *In the rebuttal, the authors empirically investigated the impact of the latent representations and argued that the chosen representation offers best generation quality (in sFID) and diversity (in Recall). However, this argument seems not sufficiently supported by the experimental results since the chosen representation was inferior to the other candidates in terms of other quality metrics such as IS, FID, Precision and the authors did not discuss why these metrics could be ignored when choosing the latent representation.*
- Unclear rationale for the hyperparameter setting (KAa1, yVZe)
	- *The authors demonstrated the impact of the hyperparameters (\alpha and \beta) on performance by additional experiments in the rebuttal. but the range of hyperparameter values examined is quite limited. More importantly, the rationale for the selection of the hyperparameter values is unclear, and the experimental results suggest that the proposed method is sensitive to the hyperparameter setting.*
- Missing discussion regarding the trade-off between tail and head classes (KAa1)
	- *The authors simply presented head and tail class performance of the proposed method and its baseline counterpart, and did not justify the high generation quality for head classes at all.*

[Concerns that have been well addressed]
- Insufficient experimental verification (KAa1, 2xLz)
	- *The reviewer KAa1 suspected that the performance gain by the proposed method might stem from training acceleration rather than an actual improvement in generation quality for tail classes. The authors successfully addressed this concern by additional experiments in the rebuttal: the proposed method outperformed the baseline even with substantially less training epochs, and was on par with the latest work mentioned by the reviewer (i.e., REPA).*
	- *The reviewer 2xLz wanted to see the isolated effect of the CCUA loss function to better evaluate the empirical and independent contribution of the proposed method. The authors addressed this concern successfully by an additional experiments in the revision.*
- Incremental novelty (yVZe)
	- *the reviewer yVZe commented that the proposed method is a direct adaptation and combination of existing ideas: dispersive losses and alignment losses for long-tailed GAN. The authors disagreed with this comment: in the initial submission, the difference between their work and the dispersive loss was discussed and they were compared empirically; the alignment loss is novel since it has not been explored for diffusion models, used in GANs though. The AC considers this concern has been sufficiently resolved.*
- Missing comparisons with latest work (KAa1, yVZe)
	- *The authors compared their method with REPA. Moreover, the reviewer yVZe did not leave sufficient details for identifying the previous work they mentioned, and did not participate in further discussion. Hence, the AC believes this issue has been addressed.*
- Evaluation and ablation study with a limited set of performance metrics (2xLz, jyaR)
	- *This concern has been well addressed by adopting diverse performance metrics, including separate FID scores for head and tail classes, for most of the additional experiments in the rebuttal.*

**Reviewer Scores:**

The only positive reviewer (2xLz) was satisfied by the rebuttal and kept their original rating (6, borderline accept). However, the AC found that the major concerns raised by the negative reviewers have not been fully resolved as summarized above, and thus expects that they would keep their original ratings (or upgrade up to borderline reject) even if the discussion phase has continued.

---

### Decision · Program_Chairs · 2026-01-26

Reject